# Proximity proteomics identifies PAK4 as a component of Afadin–Nectin junctions

Yohendran Baskaran [1,5], Felicia Pei-Ling Tay[2,5], Elsa Yuen Wai Ng[1], Claire Lee Foon Swa [3], Sheena Wee [3], Jayantha Gunaratne[3] & Edward Manser [1,4 ✉]

Human PAK4 is an ubiquitously expressed p21-activated kinase which acts downstream of Cdc42. Since PAK4 is enriched in cell-cell junctions, we probed the local protein environment around the kinase with a view to understanding its location and substrates. We report that U2OS cells expressing PAK4-BirA-GFP identify a subset of 27 PAK4-proximal proteins that are primarily cell-cell junction components. Afadin/AF6 showed the highest relative biotin labelling and links to the nectin family of homophilic junctional proteins. Reciprocally >50% of the PAK4-proximal proteins were identified by Afadin BioID. Co-precipitation experiments failed to identify junctional proteins, emphasizing the advantage of the BioID method. Mechanistically PAK4 depended on Afadin for its junctional localization, which is similar to the situation in *Drosophila*. A highly ranked PAK4-proximal protein LZTS2 was immuno-localized with Afadin at cell-cell junctions. Though PAK4 and Cdc42 are junctional, BioID analysis did not yield conventional cadherins, indicating their spatial segregation. To identify cellular PAK4 substrates we then assessed rapid changes (12') in phospho-proteome after treatment with two PAK inhibitors. Among the PAK4-proximal junctional proteins seventeen PAK4 sites were identified. We anticipate mammalian group II PAKs are selective for the Afadin/nectin sub-compartment, with a demonstrably distinct localization from tight and cadherin junctions.

[1]sGSK Group, Institute of Molecular & Cell Biology, A*STAR, Singapore, Singapore. [2]FB Laboratory, Institute of Molecular & Cell Biology, A*STAR, Singapore, Singapore. [3]Quantitative Proteomics Group, Institute of Molecular & Cell Biology, Singapore, Singapore. [4]Department of Pharmacology, National University of Singapore, Singapore, Singapore. [5]These authors contributed equally: Yohendran Baskaran, Felicia Pei-Ling Tay. This paper is dedicated to the memory of Louis Lim. ✉email: ed.manser@imb.a-star.edu.sg

Mammalian PAK isoforms are categorized into two groups on the basis of their structural and biochemical features. The group I PAKs comprise PAK1–3, which function at focal adhesions and the centrosome through their binding to the PIX/GIT1 complex[1,2]. These serine/threonine kinases also function at the plasma membrane in clathrin-independent endocytosis[3] and are implicated in the entry of several viral pathogens including HIV[4]. The group II PAKs (PAK4–6) are encoded by three genes in mammals[5–7]. PAK4 is ubiquitous[8,9] with mRNA levels highest in colon, kidney and prostate[10] consistent with a role in epithelial homoeostasis. Loss of PAK4 by genetic ablation in mice is embryonic lethal[11]. Though PAK5 or PAK6 knockout mice are viable, combined PAK5−/− PAK6−/− mice exhibit neuronal and behavioural defects[12,13]. Several studies have shown that PAK4 is oncogenic when overexpressed[14,15] and promotes tumorigenesis in vivo[16]. Amplifications of the *PAK4* gene have also been identified in pancreatic cancers[17] and the kinase can be required for proper formation of some endothelial structures[18], consistent with defects seen in *PAK4*−/− mice[19]. The development of PAK4 inhibitors has been the focus of a number of efforts[20–22], and clinical trials using PAK4 inhibitors are reported for PF-3758309 and KPT-9274.

We have shown that Cdc42 directly regulates PAK4 activity in mammalian cells through an auto-inhibitory domain (AID) that binds the catalytic domain in a manner similar to pseudo-substrates[23,24]. The regulation of PAK4 contrasts with PAK1—whose conventional activation occurs primarily through activation loop Thr-423 phosphorylation[25]. By contrast, PAK4 is constitutively phosphorylated on the equivalent activation loop site Ser-474 (ref. [26]) which may occur shortly after protein translation, as suggested for protein kinase-A (PKA). Co-expression of Cdc42(G12V) can activate PAK4 ~3-fold, when subsequently measured in vitro[26], and SH3 interaction with the PAK4 AID provides an alternative route to kinase activation[24] though in vivo demonstration for this model needs to be tested.

Vertebrates express endogenous inhibitors for PAK4 termed Inka1 and Inka2 (ref. [27]) which are primarily expressed in early development. The Inka1 protein contain two copies of a sequence which binds tightly to both the substrate-binding pocket and adjacent activation (A)-loop[23]. The evolutionarily conserved 38 residue central domain called the "Inka-box" is responsible for kinase inhibition with ~40 nM[23] affinity. In mice Inka1 and Inka2 mRNAs are primarily expressed in the developing nervous system[28,29]; these gene products likely compensate each other since loss of Inka1 gene has a very mild phenotype[29]. In *Drosophila* loss of PAK4, mushroom body tiny (Mbt), is associated defects in the fly brain[30] and some epithelia. Mbt can phosphorylate the β-catenin homologue Armadillo[31], thereby weakening cell–cell interactions[32]. More recent studies show that Mbt and Canoe (*Drosophila* Afadin) together direct *Drosophila* Par3 localization and assembly of the adherens junction (AJ)[33]. Dorsal closure requires *Drosophila* Pak1 to restore cell–cell adhesions and septate junction formation, acting through scribble during a mesenchymal-to-epithelial-like transition[34]. Thus both dPak1 and dPak4 contribute to the proper polarization in this epithelium.

In human bronchial cells[35], corneal cells[36] and other epithelia, endogenous PAK4 is enriched at cell–cell junctions. Knock-down of PAK4 does not affect collective cell migration rates but it can disrupt vertebrate cell polarization[37], which is consistent with a role downstream of Cdc42 (ref. [38]). PAK4 has also been shown to be required for spindle orientation in mitosis[39]. Other Cdc42 effectors involved in cell polarization are Par6 (ref. [40]), CIP4 (ref. [41]), DAAM1 (ref. [42]) and MRCK[43]. It is important to note that loss of cell polarity is a hallmark of many cancer cells[44,45].

The few substrates of PAK4 reported to date include β-catenin[46,47], the cofilin phosphatase SSH1 (ref. [48]), and at least two RhoA activators termed GEF-H1 (ref. [49]) and PDZ-RhoGEF[50] suggesting PAK4 modulates RhoA activation. The polarity protein Par6B, which also binds Cdc42, can be phosphorylated by PAK4 (ref. [51]). In order to find other PAK4-associated proteins we initially tried to identify PAK-binding proteins by co-purification mass spectrometry (MS); these experiments curiously yielded no specific partners. Thus, we investigated the in vivo environment around PAK4 using the BioID method first described by Roux et al.[52] modified with stable isotope labelling with amino acids in culture (SILAC) implementation[53]. These experiments demonstrated that PAK4 and Afadin share a similar cellular environment at the nanoscale which appears to include a local protein termed leucine zipper tumour suppressor 2 (LZTS2). We confirm that this Afadin/PAK4 compartment can be spatially distinct from apical TJ and more basally located cadherin-mediated junctions[54].

## Results

**PAK4 proximal proteins are predominantly junctional.** We and others found that the ubiquitous (group II) PAK4 is localized primarily at cell–cell junctions in mammalian cell lines such as MDCK, MCF7 and U2OS[35,37]. This is consistent with observations of an evolutionary conserved role with *Drosophila* PAK4/Mbt, promoting junctional polarity by regulating zonula adherens (ZA) stability though Bazooka (Par3) in the *Drosophila* eye[55]. Nonetheless the junctional compartment(s) to which PAK4 is targeted in mammalian cells has not been established, although Cdc42 plays an essential role in localizing PAK4 to cell–cell junctions in bronchial epithelial cells[35]. While the group I PAKs are targeted to specific subcellular structures such as focal adhesions and the centrosome via a specific adaptor PIX[1,56], a PAK4 'interactome' did not reveal an equivalent adaptor, nor any junctional protein among >300 proteins[57].

The BioID method with stable isotope labelling (SILAC) provides an unbiased view of the cellular environment within 20 nm of a given target protein[52,53]. This method takes advantage of the ability of BirA to label local proteins that are then detected by mass spectrometry (MS), which can be recovered on avidin Sepharose in the presence of 1% sodium dodecyl sulfate (SDS), to disrupt protein–protein interactions. Because larger N-terminal tagged constructs (cf. GFP-PAK4) interfered with proper PAK4 localization (data not shown), we generated U2OS cell-lines expressing C-terminally tagged PAK4-BirA*-GFP as illustrated in Fig. 1a. We were also able to use these PAK4-expressing cell lines to carry out standard anti-GFP immuno-precipitations in parallel. The transgene was determined to be expressed ~5-fold higher than endogenous PAK4 (Fig. 1b), but we did not observe any alteration in cell–cell junctions. This PAK4-BirA*-GFP protein was correctly localized primarily at cell–cell junctions (Fig. 1c). Addition of biotin to media allows proximal biotinylation in vivo[53], and of ~600 proteins recovered 27 yielded a SILAC ratio of >3.0 from PAK4-BirA*-GFP cells versus the GFP-BirA* control cell line (Table 1). Combining two independent SILAC-BioID MS experiments, as we described previously[53] the PAK4 dataset was tabulated according to SILAC enrichment (Table 1). The list of 27 PAK4-proximal proteins (Table 1) show these are primarily components of cell–cell junctions (cf. Afadin, ZO-1 and nectin-2); among these only p120ctn (catenin-δ1) has previously been described as a substrate of PAK4 (ref. [47]). The only transmembrane proteins were nectin-2 and nectin-3 (that bind Afadin[58]) and EphA2 that is was found in tight junctions[59]. Afadin showed significantly higher enrichment than others (SILAC ratio) suggesting it is physically closest to PAK4. Afadin

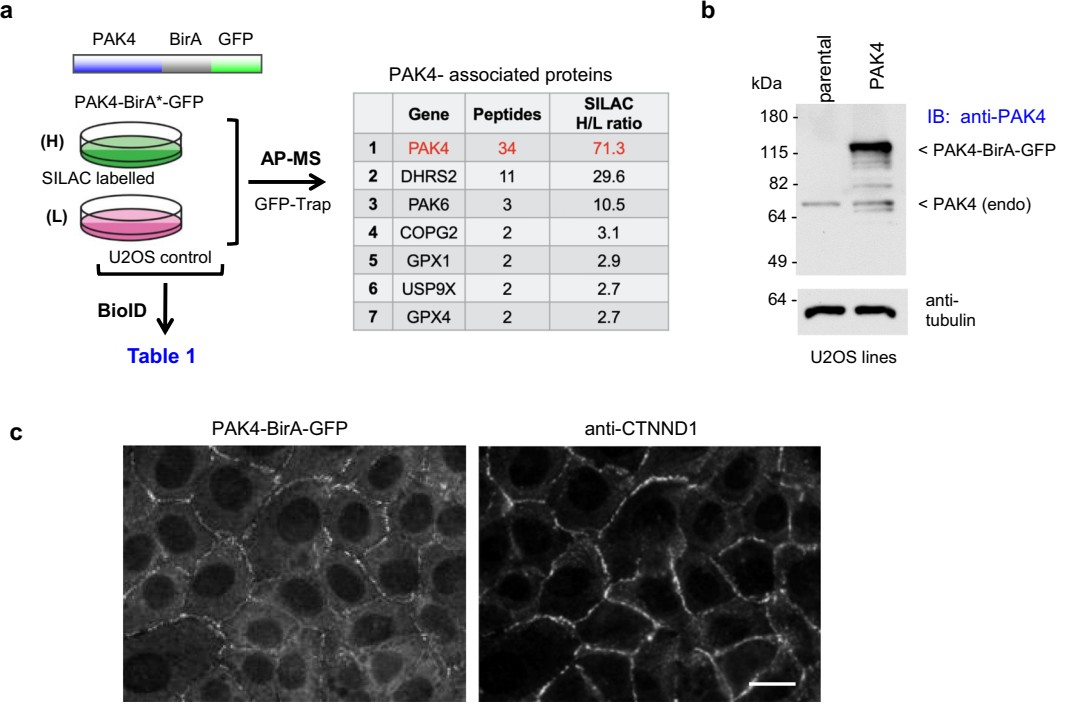

**Fig. 1 Workflow of SILAC-enriched analysis of PAK4- proximal proteins. a** Schematic of PAK4-BirA*GFP constructs used for SILAC BioID analysis. Summary of workflow used to identify PAK4-proximal proteins using stable cell lines cultured in either Arg/Lys isotopically heavy (H) or light (L) containing media as indicated. **b** Total lysate from control and PAK4-BirA*GFP cell lines were subjected to western blot to compare the expression of PAK4-BirA*GFP versus endogenous PAK4 (arrow) and repeated in two independent experiments. **c** Disposition of PAK4-BirA-GFP in stable U2OS cell lines. Cells were fixed in PFA and immuno-stained for anti-GFP and anti-p120ctn. Source data are provided as a Source Data file.

(AFDN, MLLT4) is an evolutionarily conserved protein containing two Rap1-binding domains (Fig. 2c) that bind the transmembrane nectin proteins (annotated as PVRL1–3)[58]. Afadin has been shown to directly bind a number of other junctional including p120ctn[60] and ZO-1 (ref. [61]). The appearance of both Shroom, an adaptor for ROCK, and DLG5 that directly binds Mst1/2 (ref. [62]) suggested PAK4 signalling might intersect with both ROCK and Mst1 regulated pathways. Significant levels of the Scribble protein were identified (ranked 14), which is thought of as a basal marker of cell–cell junctions.

**PAK4 purification fails to identify associated junctional proteins.** Protein–protein interaction or 'interactome' network rely heavily on large datasets in which protein associations are established through binary interaction assays such as yeast two-hybrid or other curated 'protein–protein' interaction experiments[63], for example, humanproteomemap.org. In order to probe the direct association between PAK4 and local protein(s) a GFP co-immunoprecipitation approach was used to recover putative PAK4 complex(es) from non-ionic detergent solubilized lysates (Fig. 2a). The U2OS cell lines expressing PAK4-BirA-GFP were used (cf Fig. 1) with SILAC normalization to GFP-BirA control. Although we detected a total of ~400 proteins released from the GFP-trap Sepharose beads, only two proteins showed significant SILAC enrichment (Fig. 2a). In other words essentially all Sepharose-bound proteins represent background rather than PAK4-associated proteins. The mitochondrial protein DHRS2 (ref. [64]) is likely an artefact since PAK4 did not show mitochondrial localization. We believe (low levels) of endogenous PAK6 protein does binds to PAK4 cell–cell junctions[65]. In marked contrast ~300 'PAK4-asscociated proteins' were identified from epithelial MCF7 cell lysates[57] after transient Flag-PAK4 expression, though none of these were junctional proteins. We conclude that PAK4 interaction with junctional proteins are not maintained under standard purification conditions. Thus, our study highlights that the BioID technique is uniquely able to curate PAK4-proximal proteins.

**Significant overlap between Afadin- and PAK4-proximal proteins.** Next we wanted to establish if local protein environment around PAK4 was internally consistent with that of its most proximal partner Afadin, using a similar BioID analysis[66]. The extracted dataset with specificity >2.0 (based on spectral counts in duplicate experiments) is presented in Supplementary Dataset T1. Afadin was found to be proximal to both PAK4 and PAK6 (ranked 14 and 4, respectively) while the only other S/T kinase found was the RhoA effector ROCK1. This is consistent with the ROCK adaptor Shroom being proximal to the Afadin/PAK4 complex. Figure 2b illustrates the substantial overlap between the two datasets; taking into account the multiple protein isoforms expressed in these cells, 16/27 proteins in our PAK4 BioID set were identified with high confidence by BirA-Afadin (Supplementary Dataset T1). These Afadin-proximal proteins include proteins previously found to interact directly with Afadin, such as nectin, ZO-1, EphA2, and p120ctn[59,61,67]. We did not identify Rap1, a key regulator of cell–cell junctions[68], which binds to the Afadin N-terminal located Ras-association (RA) domains (Fig. 2c). This may reflect the transient nature of the Rap1.GTP interaction. Likewise Cdc42 was not identified as PAK4-proximal, although PAK4 was detected with high confidence in Cdc42 BioID analysis, as discussed later (see Supplementary Dataset T2).

Nectins constitute a junctional (sub)compartment formed by homo-or heterophilic *trans* interaction of their extracellular regions[58] while their cytoplasmic domains are responsible for localizing of Afadin. The nectins have been observed to form punctate cell-cell adhesion clusters apposed to cadherins

**Table 1 List of SILAC-enriched PAK4-proximal proteins.**

| Rank | Protein name | Gene | Peptides | Mr (kDa) | SILAC ratio | SD | Comment | Afadin BioID |
|---|---|---|---|---|---|---|---|---|
|  | PAK4 | PAK4 | 146 | 64 | 51.0 | 3.96 | PAK4-BirA | Yes |
| 1 | Afadin/AF6 | MLLT4 | 111 | 208 | 15.1 | 2.56 | Binds nectin | Yes |
| 2 | LZTS2 | LZTS2 | 9 | 73 | 10.8 | 3.06 | Binds SPAR | Yes |
| 3 | Disks large 5 | DLG5 | 48 | 214 | 10.0 | 2.04 | Junctional | Yes |
| 4 | Coiled-coil-domain 85C | CCDC85C | 4 | 45 | 9.7 | 1.36 | Apical junction | Yes |
| 5 | Nectin 2 | PVRL2 | 4 | 58 | 8.2 | 2.63 | TM binds afadin | Yes |
| 6 | Liprin-beta-1 | PPFIBP1 | 8 | 113 | 7.9 | 1.62 | Binds Kank2 |  |
| 7 | ZO-1 | TJP1 | 60 | 190 | 6.8 | 2.33 | TJ binds afadin | Yes |
| 8 | SPAR | SIPA1L1 | 5 | 200 | 6.5 | 1.16 | RapGAP | Yes |
| 9 | Shroom2 | SHROOM2 | 10 | 176 | 5.8 | 1.46 | ROCK adaptor | Yes* |
| 10 | ASPP2 | TP53BP2 | 8 | 126 | 5.7 | 1.63 | PP1 adaptor | Yes |
| 11 | RASSF7 | RASSF7 | 4 | 40 | 5.7 | 1.48 | RAP1 binding | Yes* |
| 12 | KN motif protein 2 | KANK2 | 11 | 91 | 5.6 | 1.29 | Binds liprin |  |
| 13 | ZO-2 | TJP2 | 11 | 137 | 5.1 | 1.66 | TJ | Yes |
| 14 | Scribble | SCRIB | 34 | 178 | 5.0 | 1.60 | Multi-PDZ | Yes |
| 15 | Dock-7 | DOCK7 | 9 | 243 | 4.9 | 2.24 | Cdc42 GEF |  |
| 16 | PAR-3 like | PARD3 | 8 | 151 | 4.8 | 1.66 | Polarity complex | Yes |
| 17 | P120 catenin | CTNND1 | 19 | 108 | 4.8 | 1.28 | AJ |  |
| 18 | Nectin 3 | PVRL3 | 2 | 61 | 4.5 | 1.25 | TM binds afadin | Yes* |
| 19 | ZAP1 | ZC3HAV1 | 7 | 101 | 3.9 | 1.36 | cell–cell adhesion |  |
| 20 | Numb-like protein | NUMBL | 6 | 65 | 3.8 | 1.24 |  |  |
| 21 | LAP2 | ERBB2IP | 33 | 159 | 3.7 | 3.52 | AJ |  |
| 22 | EFR3 homologue A | EFR3A | 2 | 93 | 3.7 | 1.34 | PI4K complex |  |
| 23 | PP1 actin regulator 4 | PHACTR4 | 2 | 79 | 3.6 | 1.35 | PP1 adaptor |  |
| 24 | Disks large 1 | DLG1 | 5 | 103 | 3.6 | 1.14 | Cell adhesion | Yes* |
| 25 | SGT1 | ECD | 5 | 77 | 3.5 | 2.65 | Polarity complex |  |
| 26 | Ephrin A receptor 2 | EPHA2 | 5 | 108 | 3.4 | 1.57 | Binds Afadin |  |
| 27 | Kin of IRRE-like 1 | KIRREL | 3 | 65 | 3.3 | 1.24 | Cell adhesion |  |

List of biotin labelled proteins derived from PAK4-BirA*-GFP-expressing U2OS cells based on duplicate experiments. All proteins with SILAC ratio >3.0 and peptide number >2 are ordered by SILAC enrichment. The standard deviation (SD) was calculated over all values for redundant quantifiable peptides. Proteins or related isoforms (*) proximal to Afadin (see 'Methods') are indicated in the last column. The total list of proteins identified (~600) is given in Supplementary Dataset T2.

structures early in junction formation, followed by the recruitment of claudins and JAMs apical side[68]. The nectins have relatively short cytoplasmic domains that bind selectively to the PSD-95/Disc-large/ZO-1 (PDZ) domain of Afadin, which in turn can recruits other proteins such as ZO-1 (ref. [69]).

Using the proteins identified by both PAK4 and Afadin BioID datasets, we manually curated validated interactions (or putative interactions) with the known disposition of proteins at the plasma membrane. We included the small GTP binding proteins Rap1 and Cdc42 to anchor which are important for these sub-complexes (Fig. 2d). Among this small set of proteins there is evidence for linkage. In this context SIPA1L1 (also known as SPAR) ranked eighth in the PAK4 BioID (Table 1) is a RapGAP that is well established as complexed to 'Leucine-zipper tumour suppressor' proteins LZTS1/2 (refs. [70,71]); the LZTS1/2 are N-myristoylated proteins which are here demonstrated as enriched at cell–cell junctions. The dotted lines represent putative interactions based on functional domains, for example, Rap1 may bind to the RASSF7/8 Ras-association (RA) domain, although unlike RASFF1-6, the RASFF7/8 proteins are not components of the Hippo pathway. Interestingly, RASSF8 is linked to the protein-phosphatase 1 (PP1) adaptors ASPP1/2 (ref. [72]) which have functional Ras-association domains that likely also bind Rap1 (ref. [73]). The ASPP2 protein maintains tight junction (TJ) integrity and polarized cell architecture, perhaps through its protein phosphatase activity[74]. It was notable that PAK4 (and PAK6) are the only Cdc42 effectors we found proximal to Afadin, although many studies show PARD3 is (indirectly) linked to Cdc42.

The evolutionary conserved Scribble (Scrib) and Erbin (LAP2) were abundant among PAK4-BirA biotinylated proteins. Scrib is considered a basolateral enriched junctional protein that can form a complex both DLG5 and p120ctn[75]. Scrib is membrane-bound via N-terminal palmitoylation[76] and with Erbin and Lano plays a redundant role in establishing junctional polarity in mammalian cells[77]. Binding of Scrib to the betaPIX C terminus is well established[78,79]; however, PIX does not bind PAK4. In summary, the BioID dataset indicate PAK4 primarily overlapping that of Afadin in cell-cell junctions, thus suggesting that the proteins might be co-localized at this site[37,65]. Therefore, we went on test the junctional disposition of PAK4 versus several proximal proteins in U2OS and MDCK cells.

**PAK4 closely follows Afadin localization at cell–cell junctions.** The immuno-localization of PAK4 in relation to ZO1, β-catenin, p120ctn, Afadin, Scrib and DLG5 are presented (see Figs. 3 and 4 and Supplementary S1). In confluent MDCK cells Afadin and ZO-1 co-localized with PAK4 (Fig. 3a), but it was clear that PAK4 was not prominent in the lateral adhesion puncta which were marked by β-catenin and p120ctn. The scattered dot-like accumulation of E-cadherin punctate adhesions in the lateral membrane of MDCK cells are[80] and similar nectin-based microcluster have been described[81].

When MDCK cells were cultured in matrigel, they undergo cyst formation and the segregation of PAK4 away from cadherin junctions was most obvious with PAK4 exclusively apical (Fig. 3a, lower panel). Nonetheless the nectin–Afadin system and the cadherin–catenin system are interconnected[82], and cooperate to associate the actomyosin with AJs in polarized cells[83]. Sub-confluent MDCK cell–cell junctions tend to be slanted rather than perpendicular, and in this case we saw PAK4 staining

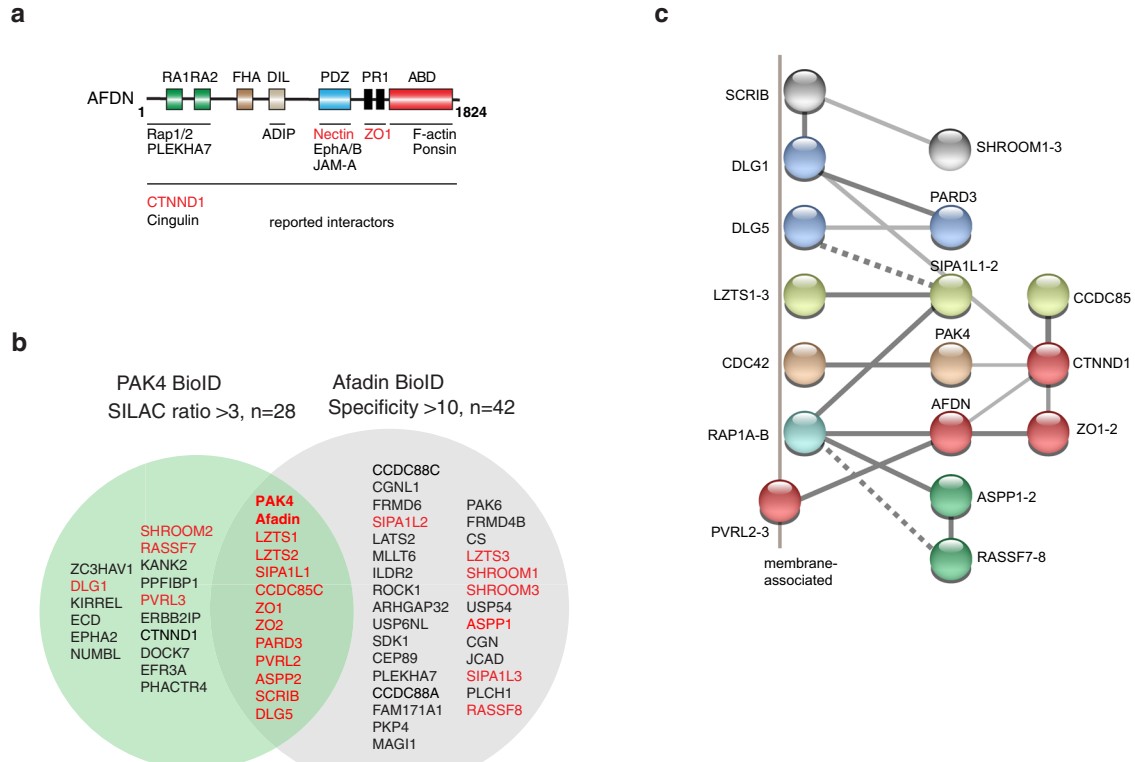

**Fig. 2 Analyses of proximal proteins identified by PAK4 and Afadin BioID. a** Schematic of Afadin domain structure and some previously identified interacting proteins, with those identified here labelled in red. **b** Diagram showing the extensive overlap between PAK4 (Table 1, SILAC with ratio >2.5) and Afadin proximal proteins (BioID data in Supplementary Data T1) which indicates these proteins share a similar junctional localization. Non-identical alternate isoforms of these shared proteins are also in red. **c** Proteins proximal to both PAK4 and Afadin are presented with putative sub-complexes (of different colours) based on validated (low throughput) protein–protein interaction data. To simplify the network different isoforms of the same proteins are shown as a single symbol, with the exception of DLG1 and DLG5. The proteins that are known to be membrane-bound (for example, by lipid anchors) or peripheral membrane proteins are placed in a juxta-membrane position.

segregated from p120 (Fig. 3b). At the edges of the MDCK colonies we often noted an accumulation of PAK4. Similarly ZO-1 segregated from and was consistently apical to PAK4 and Afadin (Fig. S3). In summary co-localization of Afadin and PAK4 was consistent in all cell types we investigated, whether in immature or mature junctions. With respect to the presence of p120ctn proximal to PAK4 (but not α-catenin or β-catenin in the BioID datasets), this is consistent with p120ctn being more widely distributed and able to associate independently with Afadin[60].

The U2OS cell–cell junctions are characterized by interdigitated structures, which are more characteristic of immature junctions[54]. By confocal imaging these structures have a brush-like appearance rather then being linear, as seen in Fig. 4. The PAK4 immuno-staining mirrored that of Afadin, while junctional β-catenin was often seen in regions that contained less PAK4 or Afadin (arrows), similar to lateral punctate junctions. We then considered whether other PAK4-proximal proteins identified here might be similarly localized. Among the 10-most proximal proteins, LZTS2 was of interest since it was found to interact with β-catenin; however N-terminally tagged proteins were not localized to junctions[84], and it was suggested that LZTS2 is cytosolic. However our immuno-localization of LZTS2 in U2OS cells showed clear staining in cell–cell junctions that co-localized with Afadin and to a lesser extent β-catenin; this antibody did not react with LZTS2 in MDCK cells. We surmise endogenous LZTS2 in U2OS cells uses an (internal) translational start site that generates an N-myristolyation signal conserved among all LZTS proteins[71]. The post-synaptic density localized LZTS proteins are

involved with maturation of dendritic spines[85]. The observation in mice that LZTS2 knockout leads to kidney defects[86] is consistent with a scaffolding role in cell–cell junctions. Thus LZTS2 behaves similarly with respect to its junctional disposition as PAK4, but further work will be needed to explore this relationship.

**PAK4 requires Afadin for its junctional localization.** Loss of Afadin is associated with temporal disruption to AJ junction formation depending on context[68]. The localization of Afadin in nectin-1 over-expressing MDCK cells was relatively insensitive to actin filament-disruption whereas other peripherally associated proteins α-catenin, vinculin, and LMO7 were lost[87]. On the other hand, Afadin accumulates at junctions upon loss of ZO-1 in parallel with enhanced F-actin and myosin IIB recruitment to junctions and was sensitive to inhibitors of acto-myosin contractility[88]. In this context, the reason that ZO-1 loss enhances the association of Shroom/ROCK at junctions is intriguing. Afadin knockdown lead to loss of PAK4 from the U2OS cell junctions while β-catenin was unaffected (Fig. S4A). Afadin knockdown did not alter either Scribble or DLG5 immuno-localization at cell–cell junctions (Fig. S1). Interestingly the loss of PAK4 was accompanied by ~50% loss of junctional Afadin relative to β-catenin (Fig. S4B), indicating that Afadin is sensitive to PAK4. Clonal MDCK cell lines depleted of Afadin show altered junctions, particularly when combined with ZO-1 loss[88]. To investigate these phenomena without the complication of clonal

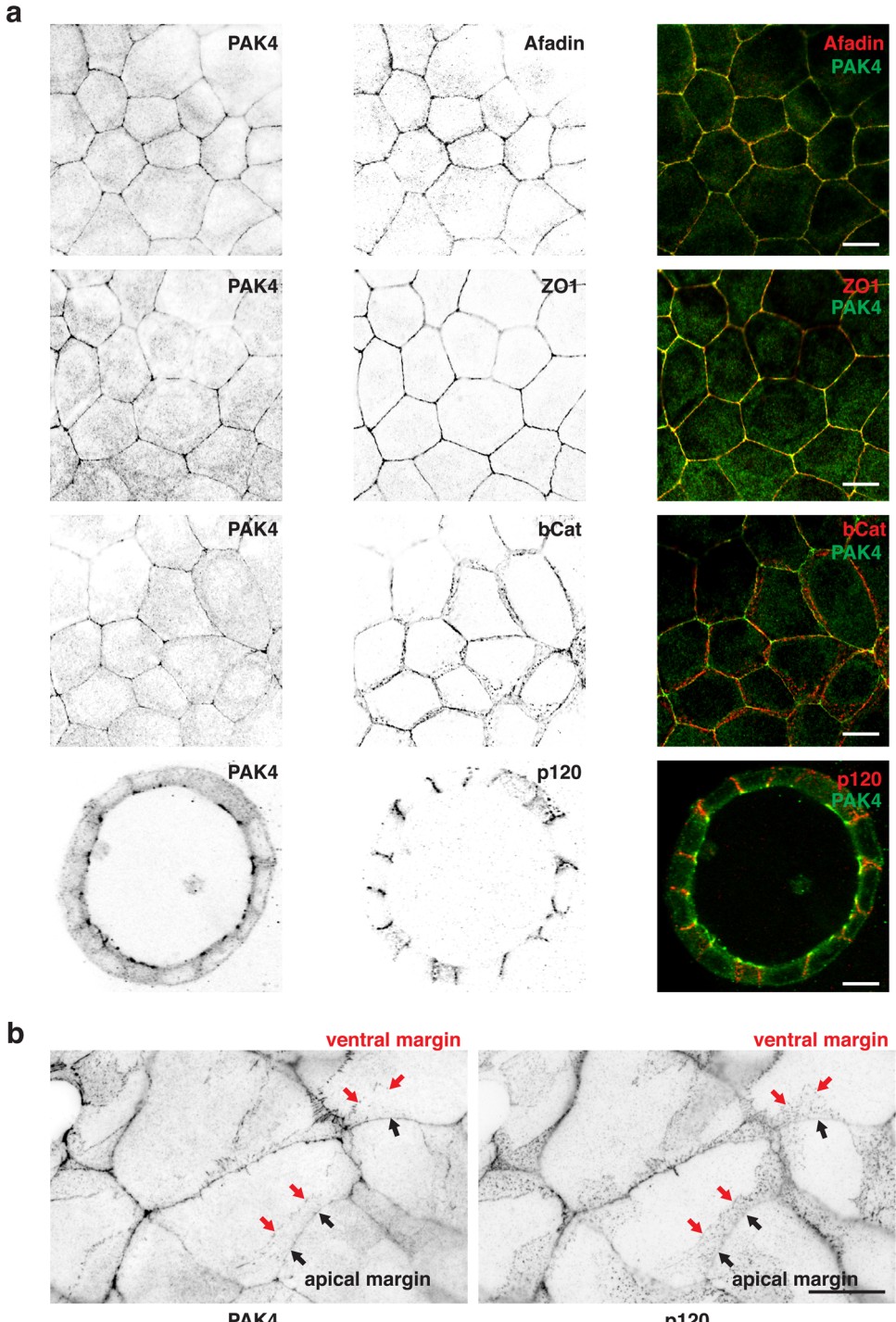

**Fig. 3 Disposition of PAK4, Afadin and ZO1 in MDCK cells. a** Confluent MDCK cells grown on glass coverslips (4 days) were fixed with methanol and co-stained using rabbit anti-PAK4 and mouse anti-Afadin/anti-β-catenin or anti-ZO1. Images were collected on an Olympus Fluoview confocal microscope with ×100 oil objective. Lower panels show MDCK cell grown in Matrigel to form 3D-cultured acini (8 days) fixed in methanol and immuno-stained with anti-PAK4, anti-p120ctn (×60 objective). **b** Sub-confluent (2 day) cells were stained using rabbit antibodies specific for PAK4 and mouse antibody specific for p120 (×60 objective). Repeated in three independent experiments. Scale bars: 10 μm.

selection, we established transient knockdown of Afadin or PAK4 in MDCK cells (Fig. 5). Cells cultured in standard media for 72 h post-transfection siRNA show ~40% mosaic loss of target expression. Under these conditions cells lacking Afadin showed a profound loss of PAK4 (Fig. 5a) while β-catenin staining was unaffected (data not shown). By contrast, PAK4 loss reduced but did not abolish Afadin levels at cell–cell junctions (Fig. 5a). We

conclude that PAK4 localization to junctions is strongly dependent on prior recruitment of Afadin, and that a feedback loop from PAK4 helps to maintain Afadin at this site.

**Inhibition of actomyosin enhances PAK4 junctional localization.** As reported two decades ago, the behaviour of E-cadherin

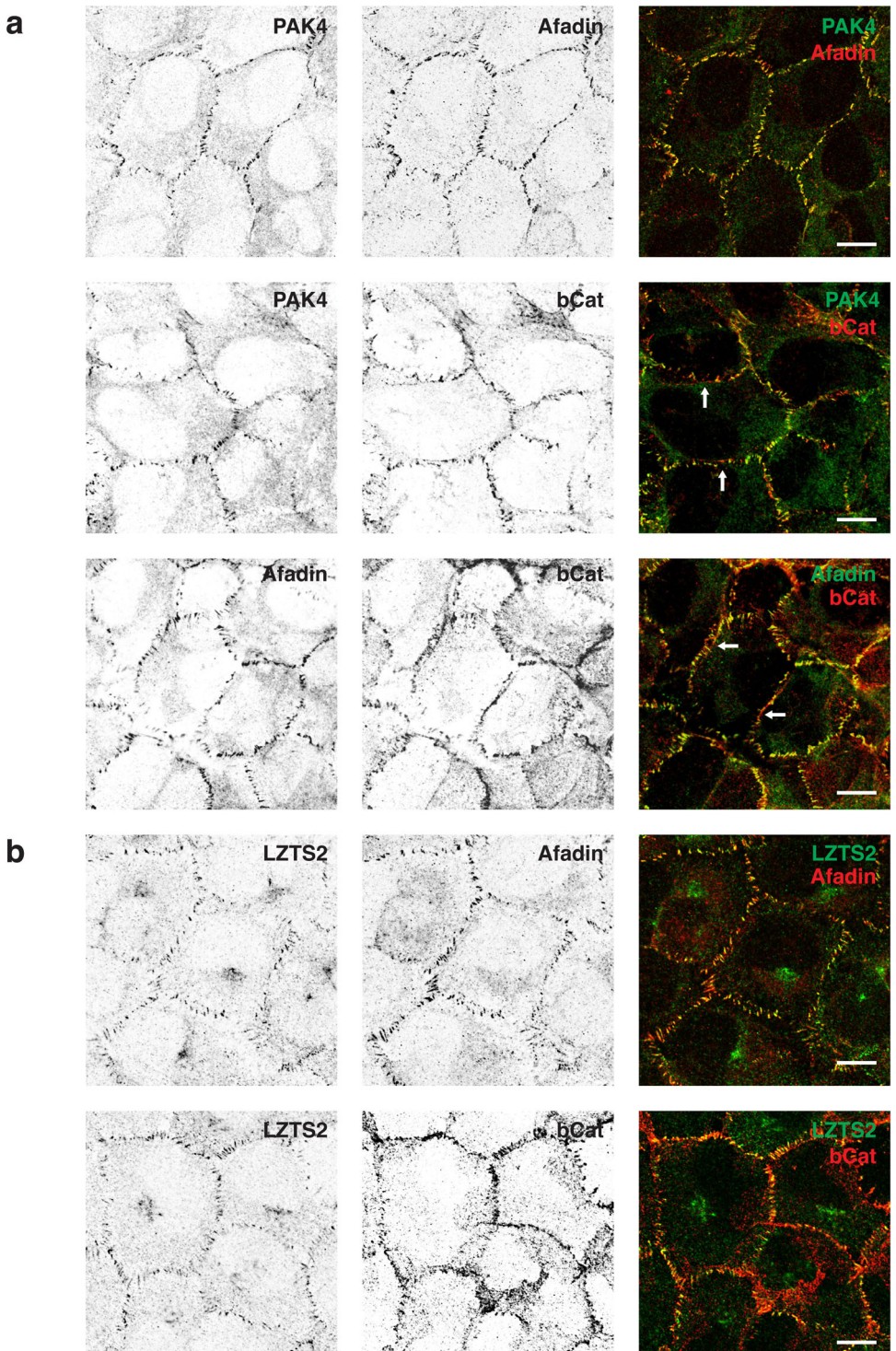

**Fig. 4 Disposition of PAK4, Afadin, LZTS2 and β-catenin in U2OS cells. a** Confluent U2OS cells grown on glass coverslips for 3 days then fixed with methanol and immuno-stained for PAK4 and Afadin or β-catenin antibodies as described in 'Methods'. Note the concordance in localization (top panel) between PAK4 and Afadin compared with that of β-catenin. Images were collected on an Olympus Fluoview confocal microscope with ×100 oil objective. **b** Cells were similarly stained using anti-LZTS2 (rabbit) and mouse anti-Afadin, or anti-β-catenin. Again note the concordance of staining of LZTS2 with Afadin. Repeated in three independent experiments. Scale bars: 10 μm.

and Afadin is quite different with respect to junctional disassembly in MDCK cells[89]. In media containing low Ca2+ rapid endocytosis of E-cadherin occurs as a result to extracellular homophilic disruption, but neither Afadin or ZO-1 compartments were disrupted[89]. This observation is evidence for an independent regulation of E-cadherin versus nectin/Afadin junctions, at least in the acute phase of cadherin internalization. Further, Par-3 is a conserved polarity protein that is needed for formation of cadherin junctions in MDCK cells, but is not necessary for assembly of homophilic nectin-based adhesions[90]. Studies have previously linked the over-expression of active PAK4 mutants to the dissolution of the acto-myosin network[14]. Such

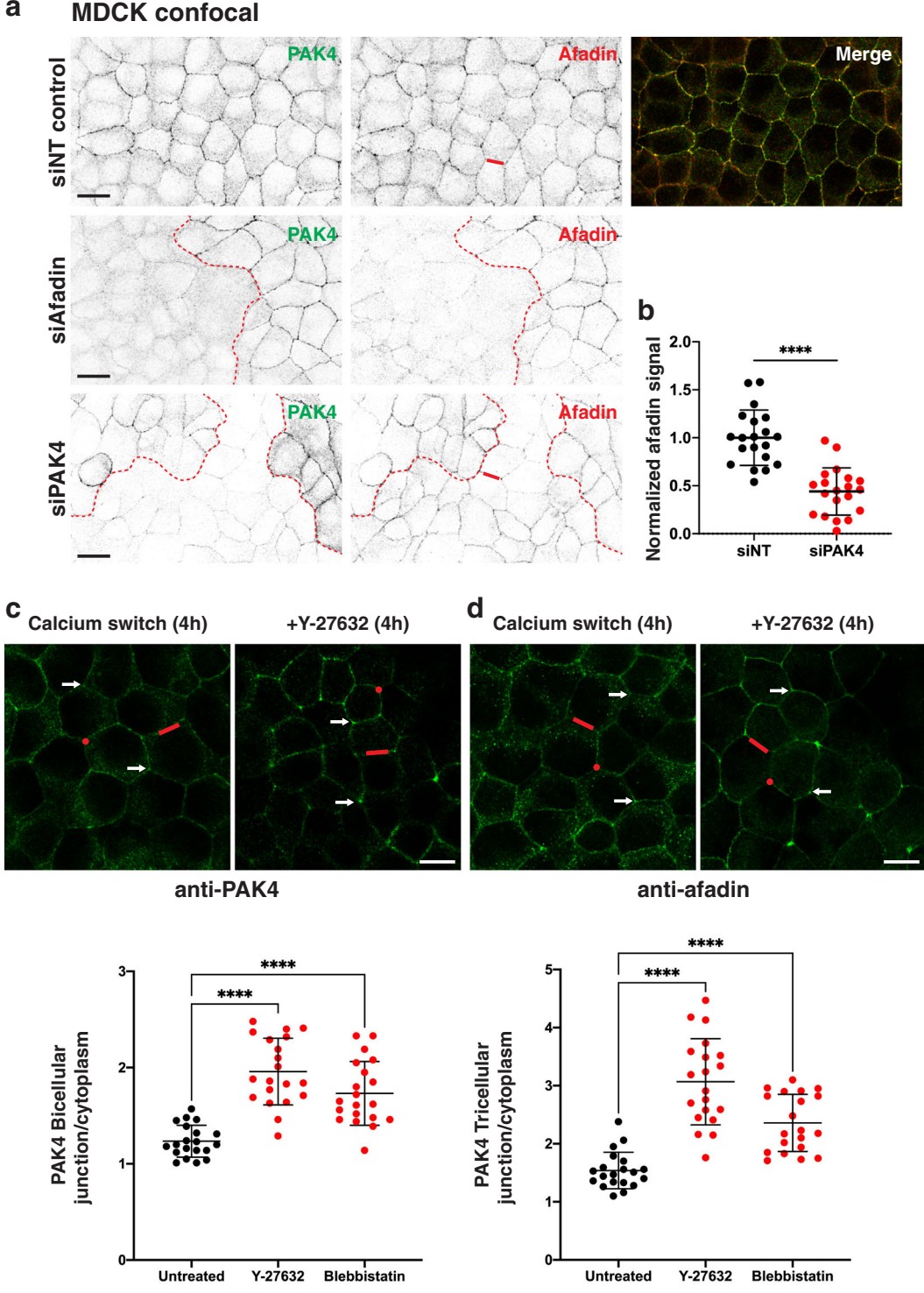

experiments are problematic because transfected cells are loaded with unregulated kinase for extended periods. In order to investigate potential feedback between the acto-myosin network and the localization of PAK4, we chemically inhibited myosin II directly or by through its regulatory kinase ROCK. Short-term (20 min) inhibition of ROCK (10 μM Y27632) is sufficient to abolish actomyosin arrays in MDCK cells with prominent loss of myosin IIB from bicellular and tricellular junctions[88]. MDCK clones lacking ZO-1 have enhanced acto-myosin, suggesting ZO-1 normally functions to suppress the local Shroom–ROCK pathway[88].

We then investigated the relationship between myosin II-based contractility and the recruitment of Afadin and PAK4 during junctional re-assembly (Fig. 5). The levels of Afadin and PAK4 were assessed by measuring the absolute relative cytoplasmic/junctional protein as detected by the relevant antibodies (Fig. 5c, d). The ratio of PAK4 or Afadin was assessed relative to β-catenin (Figs. 5c and S5), and the signal associated with bicellular or tricellular junctions versus cytoplasmic PAK4 quantified (Figs. 5d and S6). In our hands neither blebbistatin nor Y27632 significantly affected the β-catenin signal in newly formed junctions (Fig. S5). Both inhibitors significantly increased PAK4 protein levels (2–3-fold) measured at bi- and tri-

**Fig. 5 Afadin and contractibility dependent localization of PAK4 to cell–cell junctions. a** MDCK cells were treated with non-targeting (NT), Afadin or PAK4 siRNA for 72 h as indicated. Following fixation in methanol, the cells were immuno-stained for PAK4 and Afadin (Mab) as indicated. Confocal images were taken at ×100 magnification. The red-dotted line indicates the boundary of clonal knockdown cells. Repeated in three independent experiments. **b** Scatter plot showing relative protein levels derived from confocal immuno-fluorescent images. The junctional fluorescence signal was calculated for multiple 10 × 50 pixel junctional regions (cf. red bar, see 'Methods') and displayed as a ratio relative to the NT control (20 cells over 2 independent replicates). Data from two independent experiments were combined with bars indicating standard deviation from the mean analysed using a two-tailed, unpaired *t*-test (*p* < 0.0001). Scale bars: 20 μm. **c**, **d** MDCK cells were grown to 80% confluence on uncoated glass coverslips, rinsed in calcium-free PBS and incubated in 5 mM EGTA, serum-free DME for 45 min, until more then 50% cells showed rounding. Media containing 5% serum (1.8 mM calcium) was added for 45 min to allow reattachment, before addition of inhibitor/DMSO for 4 h. Cells were then fixed in methanol and immuno-stained for PAK4 or Afadin antibodies and images were taken with an Olympus Fluoview confocal microscope with ×100 oil objective. White arrows indicate enrichment at tricellular junctions. The bicellular junctional fluorescence signal was calculated as above (cf. region with red bar) and tricellular junctional fluorescence signal was calculated for a standard 15 × 15 pixel circle (red) after removal of local background (non-junction) signal. Lower panel shows scatter plots as indicated (20 cells over 2 independent replicates) from two independent experiments. Bars indicate standard deviation from the mean and analysed using an ordinary one-way ANOVA test (****p < 0.0001). Scale bars: 10 μm. Source data are provided as a Source Data file.

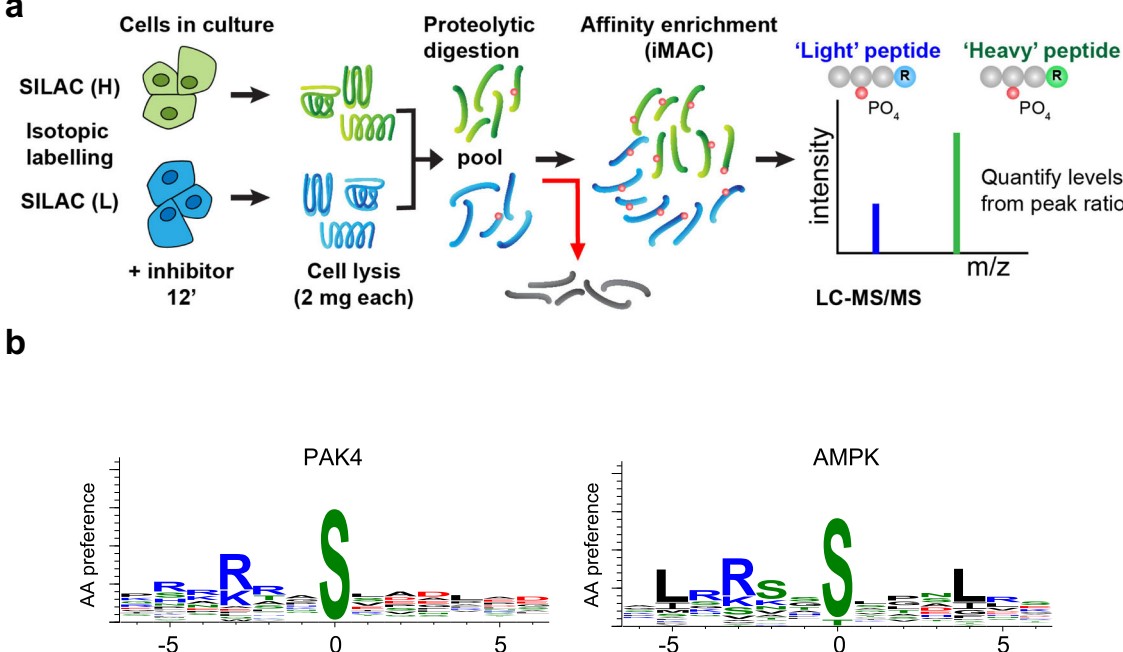

**Fig. 6 Analysis of PAK4 proximal substrates by phospho-proteomics. a** Schematic of phosphoproteomic analysis to identify phosphorylation sites that are differentially sensitive to PF3758309 versus Frax597. Summary of the workflow used to identify phospho-peptides that were depleted following PAK inhibitor treatment (12 min). The p-peptide changes were derived from isotopic ratio extracted from LC-MS/MS spectra. (experiments carried out in 'forward' and 'reverse' directions). **b** A combined matrix for the PAK4 sites was generated using WebLogo 3.7.4. as for a consensus motif for AMPK[113].

cellular junctions as indicated in Fig. 5c, d. Afadin accumulation at junctions also occurred but was less pronounced, suggesting that increased levels of junctional Afadin of itself cannot explain increased junctional PAK4.

BioID essentially represents an in vivo biochemical analysis reflecting the predominant environment experienced by the test protein[91]. Why does PAK4 or Afadin BioID not identify 'classical' adhesion markers such as E-/N-cadherin or β-catenin? Our imaging data suggest that Afadin/nectin is aligned to E-cadherin complexes in many regions (Figs. 3 and 4); however, spatial segregation is likely when assessed by BioID (radius 20 nm). Interestingly super-resolution imaging found E-cadherin and nectin-2 segregated in both epithelial A431 and DLD1 cells, as well as primary keratinocytes[54]. Current models of vertebrate epithelial junction organization[92,93] are incorporated into our schematic (Fig. 8) which highlight the key transmembrane proteins (labelled on the left) and common cytoplasmic components (right).

**Is Cdc42 enriched at Afadin/nectin junctions?** It has been suggested that Cdc42 interaction with the N-terminal CRIB domains key to PAK4 recruitment at cell–cell junctions[35]. Cdc42 is present at cell–cell junctions as well as other plasma membrane locations[94], but was not identified in the PAK4 BioID set (Table 1). In order to test if Cdc42 was present in the Afadin/nectin compartment we carried out BioID with U2OS cell lines expressing GFP-BirA-Cdc42 (wild type). The enriched dataset comprised 46 proteins with SILAC enrichment >3 (Supplementary Dataset T2). Predominantly these proteins were either transmembrane (TM) or known Cdc42 effectors (red). Among the former were multiple components of the integrin complex (Dataset T2 yellow) including MCAM and DICAM as we identified with kindlin-2 (ref. [53]), as does CD98 heavy chain SLC3A2 (ref. [95]). In addition to their presence at focal adhesions, integrins are enriched at cell–cell junctions in epithelial cells[96] and U2OS cells[53], for reasons that are clearly of interest but unresolved.

Nectin-2 (PVRL2) was Cdc42-proximal (ranked 5) but no classical cadherins were identified in spite of their abundance.

Many Cdc42 effector proteins (red text) which contain PAK-like Cdc42/Rac interaction (CRIB) domains were identified, including Borgs (Cdc42EP1/3/4), PAK2, PAK4, IRSp53 (BAIAP2), and MRCKβ (Cdc42BPB), as well as formins FMNL1/2 which binds Cdc42 through a different interface[97]. It was notable that PAK4 is the most SILAC-enriched protein kinase in U2OS cells. Prominent cell–cell junctional proteins (blue text) include Neph1 (KIRREL) which forms a complex at the podocyte intercellular junction[98]. We surmise that the critical polarity regulator Cdc42 is indeed enriched at cell–cell junctions, and proximal to several transmembrane proteins including nectins. Our dataset indicates Cdc42 is not vicinal to E-cadherin. Similar studies reveal that ubiquitous Cdc42 effectors Cdc42EP, PAK4, IRSp53, MRCK, and N-WASP were not proximal to E-cadherin[99,100]. It is notable that removal of PAK4 N-terminal basic region (residues 1–8) resulted in complete loss of junctional PAK4 and nuclear accumulation of the kinase (Supplementary Fig. S6), which is in line with results reported with PAK5 (ref. [65]). We believe this basic region adjacent to the CRIB interacts with the plasma membrane as for PAK1 (ref. [101]). It is possible that in the context of such membrane-bound PAK4 a direct Afadin interaction occurs; however, it was not possible to detect co-precipitation these proteins (data not shown).

**Identification of putative PAK4 substrates by SILAC phospho-proteomics**. Central to understanding kinase function is the identification of candidate substrates and the site(s) of specific phosphorylation. Previous efforts to characterize cellular substrates by phospho-proteomic analysis in wild type versus PAK4 −/− fibroblasts, did not identify known PAK4 substrates or new candidates[102], perhaps because of compensation due to 'network rewiring'. To date, few PAK4 substrates have been validated in vivo[9]: those identified previously by interaction assays include ARHGEF11 (ref. [50]), p120ctn[47] and SCG10 (ref. [103]). Others have been found by a candidate approaches, such as the pro-apoptotic protein Bad[104]. It should be noted that p-peptides are not uniformly detected using typical metal-enrichment protocols, and tryptic p-peptides can be too short for C18 retention, or too long to be informative for MS/MS.

It is generally agreed that the substrates of protein kinases are determined by their cellular location(s). This is exemplified by the local action of protein kinase-A (PKA) being dictated by A-kinase anchoring proteins (AKAPs), which tether the kinase in proximity to substrates[105]. Because cAMP/PKA is required to act at many sites, a variety of AKAPs have arisen. For example, AKAP1 is essential for mitochondrial phosphorylation of the dynamin-related protein 1 (Drp1) by PKA to control mitochondrial fission[106]. Cell-based experiments indicate the active catalytic subunit of PKA-AKAP18γ remains in vicinity of the substrate complex: the local radius of effective phosphorylation was estimated to be ∼16 nm[107]. This is comparable with the radius of labelling by BioID. Since the BioID method should identify proteins that are located on average within 20 nm of PAK4-BirA, one would anticipate these include the primary substrates.

Phosphorylation motif peptide arrays are useful tools to interrogate optimal kinase substrates: PAK1 and PAK4 have similar amino acid preference for basic residues N-terminal to the phosphorylated serine (−2 to −4 positions) and hydrophobic residues at +1, +2, +3 positions[108]. Having established PAK4-proximal proteins in vivo we hypothesized that PAK4 candidate sites could be identified through their sensitivity to PAK inhibitor PF-3758309 versus Frax597. The drug PF-3758309 is a pan-PAK inhibitor with nanomolar affinity that potently inhibit both group I and group II PAKs in vivo[21,109]. Our experimental scheme

(Fig. 6a) predicates on phosphorylation sites being sensitive to acute kinase inhibition (12 min). To our knowledge, such an approach to avoid 'secondary' phosphorylation events or network rewiring is not reported. This strategy also avoids having to compensate for changes in cellular protein levels. By contrast, sustained removal of protein kinases by gene knockout (for example, AMPKs) leads to substantial 'off target' alterations in protein phosphorylation[102,110].

We first validated this approach by following PAK2 auto-phosphorylation at pSer141 and pSer197 (ref. [111]). Cellular PAK2 pSer141 and pSer197 levels (and equivalent Pak1 sites) were similarly sensitive to acute treatment with both inhibitors (Table 2). We also identified PAK2 Ser152 as a site that is affected by PAK inhibition. By contrast, no phosphorylation sites on PAK4 were sensitive to PF-3758309 consistent with the key PAK4 activation loop residue Ser474 being constitutively phosphorylated, as we reported[26]. The other sites, namely PAK4 pSer99, pSer181 and pThr207 in the non-catalytic domain, were largely unaffected by PF-3758309 treatment (Table 2), suggesting they are not dynamic auto-phosphorylation sites. We did observe inhibition of Bad pSer112 (ref. [104]) and a weak inhibition of GEF-H1 pSer885 levels (Table 2). In addition, previously described partner ARHGEF11 showed changes at pSer S271 (Table 2), RRQGpSDAAV after inhibitor treatment (Dataset T3). The first report of PF-3758309 indicated a $K_i$ of 19 nM in vitro towards PAK4 (ref. [21]) similar to that of PAK1 (reported $K_i = 14$ nM) with AMPK being a major 'off target' kinase ($K_i = 40$ nM)[21]. Frax597 however has no activity towards PAK4 nor AMPK[109]. Consistent with the above, AMPK inhibition by PF-3758309 under our conditions affected many established AMPK substrates, including ACC1, BAIAP2 (IRSp53), KLC4 and TBC1D1 (Table 2).

We next extracted SILAC informative phospho-proteomic data relating to PAK4 and Afadin proximal proteins (Table 2) to establish candidate PAK4 substrates. Except for Scribble Ser1378, none of these sites were also affected by Frax597: Scribble binds tightly to the protein βPIX which bind tightly to group I PAKs[78]. Although the dataset contains six phospho-peptides derived from Afadin (Supplementary Dataset T3), none of these were sensitive to PF-3758309, indicating that PAK4 does not target Afadin, at least at the sites identified here. However, we found 17 PAK4 candidate phosphorylation sites for junctional proteins in the PAK4 or Afadin compartment as defined by BioID. It is important to note that this list does not include AMPK sites based on large-scale studies[112,113]. Although the motif matrix generated by this PAK4 set (Fig. 6b) includes the RRxS motif common for multiple 'basic-directed' kinases, and in consensus peptide array experiments[102], it did not identify Leu+4 and Leu/Met at −5 typical among AMPK substrates. Also of note among the putative PAK4 sites residues C-terminal (+1, +2, +3) to the phospho-Ser are commonly acidic or hydrophobic (but not basic) consistent with previous studies using synthetic peptides[102]. We previously showed that expression of active Cdc42 affected the maturity of cell–cell junctions and that PF-3758309 reduced β-catenin Ser675 phosphorylation in U2OS cells[37]; however, this site was not identified among the ∼8000 SILAC informative phospho-peptides. In a similar experiment in U2OS cells, we initiated the re-assembly of junctions for 5 h after calcium depletion and observed that PF-3758309 enhanced the development of cell–cell junctions with higher PAK4/Afadin content compared to p120ctn (Fig. S2). However, given the reported role of AMPK on cell–cell junctions[114,115], it is not possible to ascribe this solely to PAK4 inhibition.

Many PAK4-proximal substrates (Afadin, Scrib, ZO-1, DLG5 and p120ctn) are heavily phosphorylated (Dataset T3). To further explore the candidate PAK4 proximal substrates phospho-serine

**Table 2 PAK4 proximal substrates identified by phospho-proteomics.**

| Gene names | Site sequence | Position | PF3758309 treated | | Frax597 treated | | Comments |
|---|---|---|---|---|---|---|---|
| **PAK phosphorylation sites** | | Phosphosite | **Fold-inhib** | **SEM** | **Fold-inhib** | **SEM** | |
| PAK2 | VKQKYLSFTPPEK | S141 | 5.25 | 1.37 | 4.95 | 1.43 | auto-phos (KID) |
| PAK2 | EKDGFPSGTPALN | S152 | 2.58 | 0.07 | 3.09 | 1.64 | |
| PAK2 | KSIYTRSVIDPVP | S197 | 2.56 | 0.3 | 1.59 | 0.35 | auto-phos |
| PAK4 | SVTRSNSLRRDSP | S99 | 0.42 | 0.03 | 0.8 | 0.04 | 14-3-3 site |
| PAK4 | RDKRPLSGPDVGT | S181 | 1.43 | 0.11 | 1.15 | 0.07 | 14-3-3 site |
| PAK4 | AGRPFNTYPRADT | T207 | 0.91 | 0.07 | 0.98 | 0.07 | |
| PAK4 | EVPRRKSLVGTPY | S474 | 1.01 | 0.08 | 0.96 | 0.05 | auto-phos (A-loop) |
| **Published PAK4 substrates** | | | | | | | |
| ARHGEF11 | HHRRQGSDAAVPS | S271 | 2.16 | 0.17 | 1.15 | | Ref. [50] |
| GEFH1 | VDPRRRSLPAGDA | S885 | 1.73 | 0.41 | 1.37 | 0.24 | Refs. [49, 147] |
| BAD | IRSRHSSYPAGTE | S75 (112) | 2.82 | 0.18 | 1.33 | 0.18 | Ref. [104] |
| BAD | FRGRSRSAPPNLW | S99 (136) | 1.36 | 0.23 | 1.17 | 0.06 | Ref. [148] AKT/PKB |
| BAD | RELRRMSDEFVDS | S118 | 1.03 | 0.02 | 1.03 | 0.1 | Ref. [149] PKA |
| **PAK4 candidate sites (this study)** | | | **SILAC >2.0** | | | | |
| CGN | PSNRSNSMLELAP | S149 | 4.69 | 1.42 | 1.22 | 0.22 | cingulin |
| CTNND1 | PRRRLRSYEDMIG | S320 | 8.16 | | 1.14 | 0.5 | p120ctn |
| CTNND1 | AQHERGSLASLDS | S346 | 7.2 | | 1.17 | 0.26 | p120ctn |
| CTNND1 | EGYRAPSRQDVYG | S252 | 3.46 | 0.55 | 1.18 | 0.15 | p120ctn |
| DLG5 | SSARLGSSSNLQF | S1263 | 4.79 | 0.67 | 1.13 | 0.13 | |
| ERBB2IP | IPERTMSVSDFNY | S1158 | 4.55 | 1.27 | 1 | 0.37 | LAP2 |
| ERBB2IP | LSARTYSIDGPNA | S1133 | 2.6 | 0.89 | 1.2 | 0.11 | LAP2 |
| KANK2 | PRERVPSVAEAPQ | S540 | 4.78 | 0.53 | 1.62 | 0.07 | |
| NUMB | GHRRTPSEADRWL | S438 | 4.81 | 1.13 | 1.04 | 0.15 | |
| PPFIBP1 | SNKRTASAPNLAE | S540 | 5.29 | 0.15 | 1.24 | | Liprin |
| PPFIBP1 | KLRRSQSTTFNPD | S601 | 4.37 | 0.42 | 0.88 | 0.1 | Liprin |
| SCRIB | GPPKRVSLVGADD | S1378 | 3.67 | 0.48 | 2.06 | 0.21 | |
| SHROOM1 | THPRSASLSHPGG | S188 | 2.87 | 0.67 | ND | | |
| SHROOM1 | QRKWCFSEPGKLD | S224 | 2.53 | 0.27 | 1.09 | 0.01 | |
| TJP1 | KRNLRKSREDLSA | S617 | 8.62 | 2.9 | 1.05 | 0.16 | ZO-1 |
| TJP2 | QMRRAASSDQLRD | S978 | 6.99 | 2.37 | 1.02 | 0.08 | ZO-2 |
| **AMPK substrates** | | | | | | | |
| ACC1 | HIRSSMSGLHLVK | S80 | 11.33 | 1.51 | 1.14 | 0.23 | Ref. [113] |
| BAIAP2 | TLPRSSSMAAGLE | S366 | 3.02 | 0.41 | 1.15 | 0.09 | " |
| KLC4 | NMKRAASLNYLNQ | S590 | 2.98 | 0.25 | 1.08 | 0.04 | " |
| TBC1D1 | PMRKSFSQPGLRS | S237 | 8.25 | 0.65 | 1.11 | 0.13 | " |
| TSC2 | PLSKSSSSPELQT | S1387 | 1.55 | 0.39 | 0.87 | 0.05 | Ref. [112] |

List of peptides identified by LC-MS/MS based on changes in SILAC ratio (shown in bold) between drug-treated and controls (average of three experiments with standard error of mean, SEM). The top panel shows PAK2 and PAK4 phosphorylation sites here identified. While two sites on PAK2 were robustly affected by both inhibitors, the PAK4 phospho-Ser181 level showed limited sensitivity to PF3758309 (1.4-fold decrease) at this time point. The blue panel shows some sites identified in proteins previously reported as PAK4 substrates. The green panel are candidate PAK4 sites with SILAC ratio >2.0 (in red) among proteins that were proximal to PAK4 or Afadin. The lower panel in grey indicate some of the sites previously identified as AMPK substrates which are referenced on the right.

peptides with basic residues (at −2, −3, −4) were extracted from the phosphosite database and in vitro phosphorylation was performed on synthetic peptides as illustrated in Fig. 7b. The typical in situ data with alanine scan substitution of a synthetic substrate SARRPKSLVDPAD, based on the PAK4 auto-inhibitory (AID) sequence with $K_d$ of 29 μM[23]. The structural basis for the binding of the AID to the catalytic domain is understood[24]. The Arg −3 interacts prominently with an acidic patch on the kinase surface[24] while residues C-terminal to the target (Ser0) form a beta-strand with the kinase activation loop. Hydrophobic side chains at +1 and +2 interact with the associated hydrophobic shoulder formed by the C-lobe. Alanine scanning indicated Arg(−3) was the only residue essential for peptide phosphorylation (Fig. 7a). Nonetheless substitution of Arg(−4), Pro (−2) and Lys(−1) did have significant impact on phosphorylation efficiency. In line with structural considerations the Leu(+1) sidechain was most sensitive to alanine substitution, while positions V(+2) and D(+3) were not.

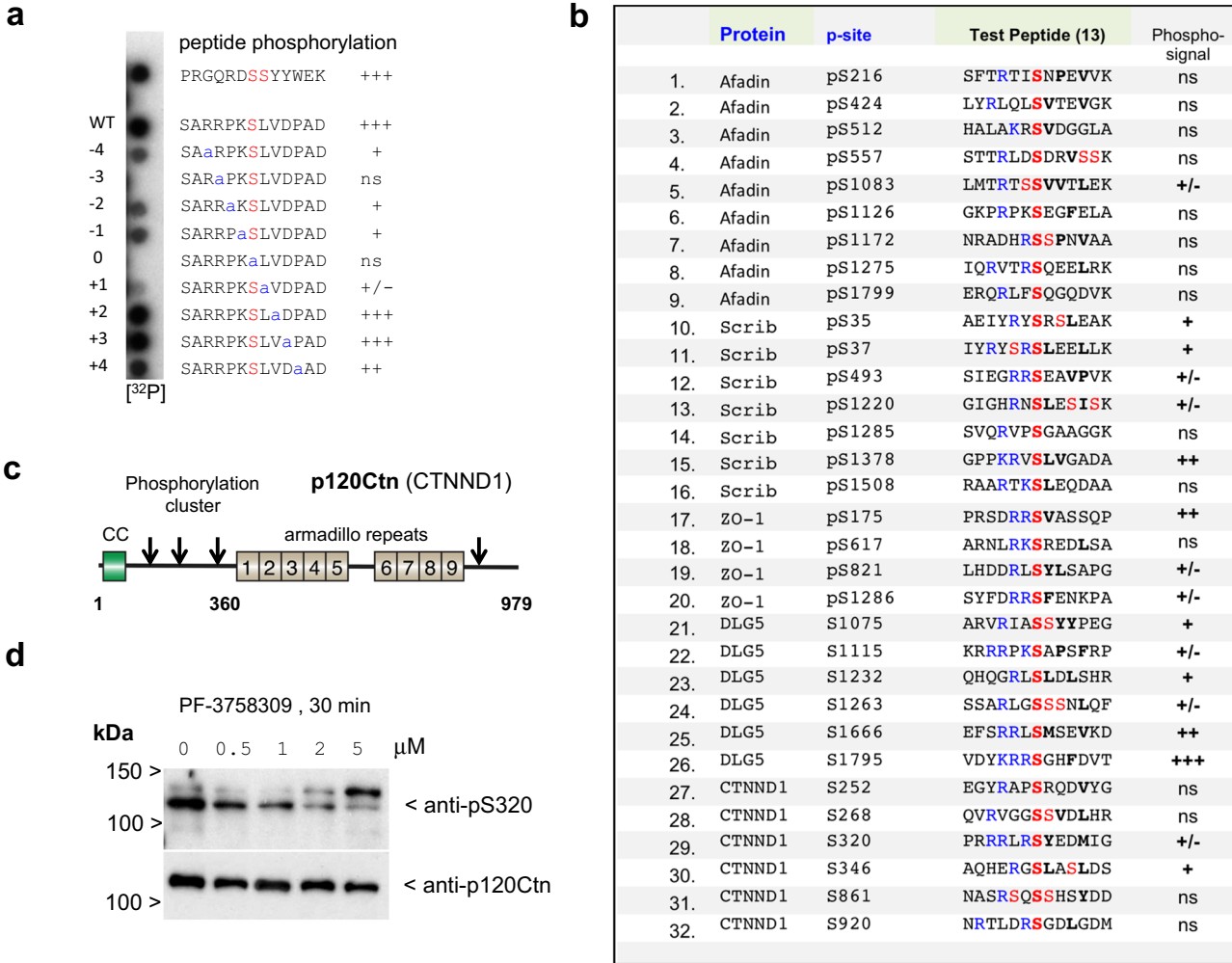

**Fig. 7 Peptide spot analysis of putative target sites in PAK4-proximal proteins. a** To assess the effect of single amino-acid substitution on a selected optimal PAK4 substrate, we tested 13aa synthetic peptides (Pepspots, Jerini) derived from PAK4 pseudosubstrate motif (SARRPKPLVDPAD) in which the proline in bold is replaced by Ser(0). This is similar to an optimal substrate for PAKs (RKRRNSLAYKK) termed PAKtide[145] but optimal for kinase binding. Based on structural considerations the Arg side chain at position −2 or −3 occupies a pocket that mediates interactions found in PAKs and other S/T kinases, including PKA[146]. The contribution of each side-chain to peptide phosphorylation was assessed by sequential alanine substitution. **b** We selected in vivo basic-directed phosphorylation sites identified in Afadin, scribble, ZO-1, DLG5 and p120ctn as compiled in the Phosphosite database (V6.5.9.3). The corresponding synthetic peptides were synthesized and subjected to in situ phosphorylation. The extent of phosphorylation (32P signal) ranges from detectable (−/+) to very strong (+++), with no signal shown as ns. **c** Schematic of the domain structure and relative positions of the p120-catenin phosphorylation sites as indicated in the table. **d** Western blot showing the inhibition of p120ctn Ser320 phosphorylation by PF-3758309 U2OS cells[113].

Of the 32 synthetic peptides corresponding to in vivo basic-directed sites of phosphorylation (Fig. 7b), 17 showed detectable levels of in vitro phosphorylation. The most highly phosphory-lated peptide, derived from DLG5 (VDYKRRSGHFDVT), does not contain hydrophobic residues at +1/+2 suggesting that basic amino-acids at −1/−2/−3 alone drive this preference. These in vitro substrates in the main showed no consensus with respect to the optimal position of the basic residue N-terminal to the phospho-acceptor, but most included a hydrophobic residue at +1 position (Fig. 7b). These findings attest to the notion that strongly phosphorylated peptides (or proteins) in vitro may not reflect in vivo targets. We next looked at the Ser320 p120ctn site, which is very sensitive to PF-3758309. The residues around Ser320 are conserved across vertebrate evolution, and a phospho-specific antibody showed inhibition by PF-3758309 in a concentration-dependent manner (Fig. 7d). This result indicates that PAK4 phosphorylates at least two sites on p120ctn namely Ser288 (ref. [47]) and Ser320, though the former site was not seen

after enrichment and MS. Between these sites (T310) was reported to be a substrate of GSK3β[116]. The phosphorylation in this region of p120ctn is known to decrease the homophilic binding affinity of E-cadherin[117]. This is interesting because p120ctn is a signalling protein whose over-expression causes Rac and Cdc42 activation[118], which is pertinent to cancer progression upon E-cadherin loss[119]. Thus PAK4 could provide a feedback loop from Cdc42 back to p120ctn through phosphorylation of Ser288 and Ser320.

## Discussion

### PAK4 and Afadin define a sub-compartment of cell–cell junctions. Cadherins have long established roles in cell–cell adhesion in the multicellular organisms, but nectin-mediated cell adhesions were discovered later[120]. Vertebrate nectins have short intracellular domains that recruit Afadin, which in turn allows their proper junctional location[69,121]. Nectins and Afadin play

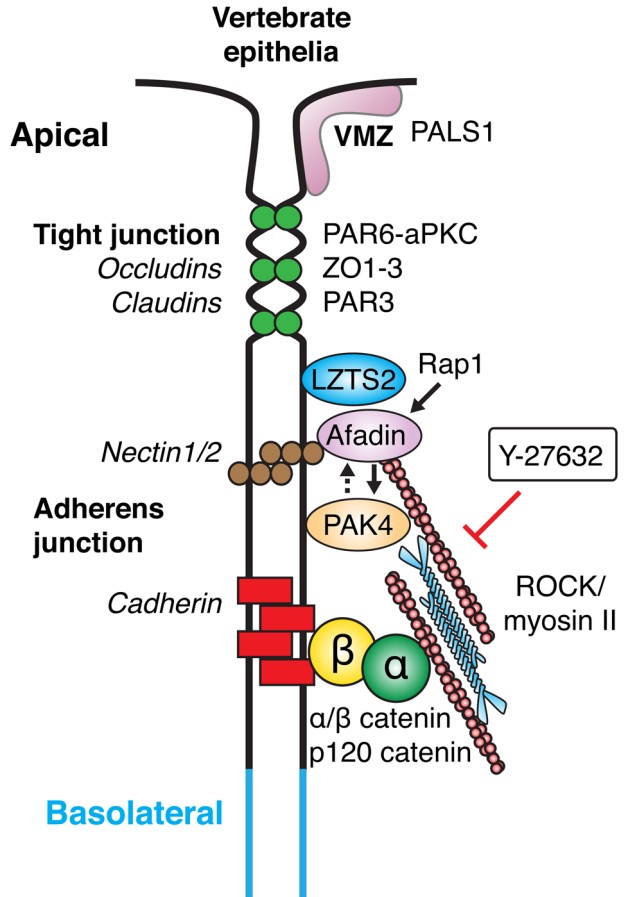

**Fig. 8 Simplified schemes showing the organization of the junctional complexes of polarized epithelial cells in vertebrates.** The sub-apical complex includes a structure recently described[126] as well the tight junction (TJ) and region 'below' this typically described adherens junction (AJ), which includes both Afadin and cadherin complexes. The smaller cadherin punctate junctions along the lateral contacts are not explicitly indicated. Typical non-transmembrane components (for example, p120ctn and β-catenin) are often used as markers and these are indicative of well-studied components. The Afadin/nectin compartment in vertebrates is often spatially segregated from cadherin as described in the text.

roles in the formation of cell–cell junctions cooperatively or independently of cadherins depending on cell type[68]. For instance, heterotypic interactions between nectin-2 and nectin-3 are required during spermatogenesis[122]. In vertebrate epithelial cells, TJs are always formed at the apical side of AJs due to cell polarization along the apical–basal axis at the cell–cell interface. Nectins may mediate the formation of TJs at this position by promoting claudins and occludin accumulation at this site[58]. It is interesting to note that nectins are involved with activation of the key polarity regulator Cdc42 (ref. [123]). In addition to nectin binding, Afadin contains two Ras-binding domains located in the amino-terminal half of the protein (Fig. 2a), which bind the small GTPase Rap1. In both vertebrates and invertebrates Rap1 activation is essential to the assembly of cell–cell junctions, as well as cell–matrix adhesions[124]. Our work here emphasizes the finding that the Afadin compartment is biochemically distinct from the cadherin one (Fig. 8), and is corroborated by optical super-resolution imaging[125]. Though a biochemical interaction between Afadin and PAK4 was not found, and it will be important to investigate other underlying protein–protein interactions responsible to keep the components of the afadin junctional

compartment segregated. For example, we are currently characterizing the interactions of LZTS2 with other proteins of the mammalian afadin complex.

This work complements a recent study[126] which uses APEX2 to develop a proteome of the apical-lateral border using both proximity analysis and electron microscopy. The Crumbs complex (Pals1, PatJ, Lin7c, and Crumbs3) defined a spatially distinct compartment apical of TJ, coined the vertebrate marginal zone (VMZ), as indicated in Fig. 8.

**Loss of PAK4 leads to junctional phenotypes.** Systemic knockout of PAK4 in mice results in embryonic lethality around embryonic day 12 (ref. [11]). This results from foetal heart defects and abnormalities in extra-embryonic tissue placenta and its vasculature. Epiblast-specific deletion of PAK4 indicates the observed extra-embryonic defects are not responsible for the major embryonic abnormalities in the heart and brain[127]. The conditional deletion of PAK4 in the brain of mice results in overall growth retardation, premature death and significantly impaired neurogenesis, likely as a result of AJ disruption in neuroepithelia[128]. PAK4 was found to promote TJ and AJ maturation in human bronchial cells[35] but knockdown did not block formation of these structures. Taken together these observations are consistent with a role for PAK4 in regulating aspects of vertebrate cell–cell junction formation.

The direct association of Afadin with the TJ protein ZO-1, which binds JAMs, occludin and claudins[69], is puzzling. One hypothesis is that prior to formation of TJs, ZO-1 transiently interacts with Afadin to drive this process[69]. On the other hand, loss of ZO-1 from MDCK cells leads to increased recruitment of Afadin to junctions, perhaps as a result of enhanced acto-myosin contractility via ROCK[88]. However, we note that Afadin recruitment is, if anything, increased when junctions are reassembled while acto-myosin is inhibited (Fig. 5d). Studies involving several cell types and tissues indicate Afadin usually occupies a location that is distinct from TJ and in some instances cadherin-based junctions (for example, Fig. 3a). By using super-resolution microscopy and live imaging, the sub-junctional distribution of cadherin and nectin in A431 cells and human keratinocytes showed that the two occupy separate clusters[54]. The size of these adjacent clusters is independent and can significantly fluctuate over time. A431 cells lacking Afadin exhibit no abnormalities in junctional morphology, but there are defects in the assembly of new AJs upon cell–cell contact[125]. This may relate to Afadin–p120ctn interactions regulating rates of E-cadherin endocytosis[129]. Our observation that PAK4 is co-localized in MDCK cells with Afadin, but less so with cadherins (Fig. 3b), particularly in puntate function, is consistent with the implied nanoscale separation of nectin/Afadin junctions in these cells.

**Conserved association of invertebrate PAK4 and Afadin.** Invertebrate model systems have been useful in providing general principles of signalling pathways and cellular organization—however it should be appreciated that in the context of cell–cell junctions fundamental differences exist in the organization of these structures[92]. In fly epithelia the ZA, which is equivalent to the vertebrate AJ, does not have an associated apical TJ, but rather the barrier function is carried out by the septate junctions located basally[92]. In the fly embryo, Rap1 and Afadin (Canoe) regulate the apical localization of both PAR3 (Baz) and β-catenin (Arm), with Baz reciprocally influencing Canoe (Cno) localization[68,130,131]. *Drosophila* Afadin and PAK4 localization are inter-dependent[33], and they regulate epithelial morphogenesis through retention of Bazooka (PAR3) at the ZA. In this context the localization of Rap1, Cno and Mbt were found to be

interdependent. As a result, PAK4 (Mbt) is required for proper ZA morphogenesis in the eye[31]. Mbt can phosphorylate β-cat/Arm and this phosphorylation was shown to limit association of Arm with E-cadherin[132]. Thus, PAK4/Mbt is required for proper epithelial junction formation through the ZA. By contrast we suggest that mammalian cells required Afadin for PAK4 junctional recruitment but not vice versa (Fig. 8). The evolutionarily conserved features of PAK4 include an N-terminal membrane-binding region[65], the Cdc42 binding (CRIB) the AIDs which regulates the catalytic domain[26,133].

The Rap1 GTPase and its 'effector' Canoe (Cno/AF6) which is the Afadin orthologue[124,134,135] functions with the Par complex cf. Cdc42–Par6–aPKC–Bazooka and Mbt/PAK4 (refs. [33,35,55]). Cno and Mbt are proposed to stabilize the E-Cadherin and Bazooka retention at the ZA. We show here that Afadin and PAK4 were not required for formation of cadherin-based junctions in vertebrate cells in culture, consistent with previous observations[125]. Thus Afadin is essential for ZA formation in flies but plays a more nuanced role in vertebrate cadherin-based junctions in epithelia[68].

Although we have not specifically tested other human group II PAKs, it is highly likely that PAK5 and PAK6, which are localized to cell–cell junctions[65], are present in the Afadin sub-complex. In the case of PAK6 this localization was shown to be dependent on both Cdc42 interaction (CRIB) and the adjacent polybasic region, which are essentially identical among the three human isoforms[65]. However, PAK5 and PAK6 differ from PAK4 in that their basal activity is significantly higher due to their oligomeric state[136]. PAK5/6 are largely restricted to the nervous system[137], and cells in culture have almost undetectable levels of the protein compared to PAK4 (ref. [136]). The over-expression of PAK6 can promote the disassembly of cell–cell adhesions[138], which underlies epithelial to mesenchymal transition, one of the hallmarks of cancer.

**Is there a distinct Afadin/nectin compartment?** Our findings offer unexpected insight to both Afadin and PAK4 biology, as the biochemical and cellular studies indicate this S/T kinase is selective for the Afadin/nectin junctional sub-compartment. Further work on LZTS2 will be required to determine how this protein is connected to the local complex, but the preliminary data (Fig. 4b) suggest a similar disposition to Afadin. The fact that the LZTS family of proteins are membrane linked through an N-myristolyation sequence[139] indicates a plasma membrane complex likely with the RapGAP protein SIPA1L1 (ref. [71]), as detected by BioID (Table 1). We anticipate that among the proteins identified as PAK4-proximal, additional protein components of the Afadin/nectin being uncovered. It will be of significant interest to investigate the single orthologues in *Drosophila* where genetic manipulation is simpler. In the context of epithelial tissues in mice, the inter-dependence of PAK4 and Afadin/nectin with respect to knockout animals would warrant further analysis.

## Methods

**Plasmids, antibodies and reagents**. Primary antibodies were obtained from the following sources: rabbit PAK4 (Proteintech 14685-1-AP); mouse p120-catenin (Santa Cruz sc-23873); rabbit β-catenin (Cell Signaling #9582S); mouse β-catenin (Santa Cruz sc-7963); rabbit Afadin (Sigma A0224); mouse Afadin (R&D Systems MAB78291); rat ZO-1 (Santa Cruz sc-33725); rabbit LZTS2 (Proteintech 15677-1-AP); rabbit DLG5 (Abcam ab86783); rabbit Scribble (Santa Cruz sc-28737); rabbit p120-catenin pS320 (Cell Signaling #8016S), mouse β-tubulin (Sigma T4026). Goat anti-Rabbit Alexa Fluor 488 (Invitrogen A-11008); Donkey anti-Mouse Alexa Fluor 546 (Invitrogen A-10036); Goat anti-Rat Alexa Fluor 568 (Invitrogen A-11077); Rabbit anti-Mouse HRP (Dako P0260); Goat anti-Rabbit HRP (Dako P0448). For immunofluorescence, primary antibodies (0.5 mg/ml) were diluted 1:100 while secondary antibodies were diluted 1:200. The PAK4 inhibitor PF-3758309 was a gift from Dr. Chernoff, Fox Chase. Blebbistatin, Y-27632 and Frax597 was obtained from Selleckchem. The SILAC medium and dialysed foetal bovine serum (FBS) was

from Thermo Scientific. The heavy L-lysine (U13C6; U15N2 K8), L-arginine (U13C6; U15N4 R10) and light L-lysine (K0) and L-arginine (R0) were from Cambridge Isotope Laboratories. High capacity Neutravidin agarose resin (cat #29204) was from Thermo Scientific. The GFP-trap_A agarose was from ChromoTek GmbH. D-Biotin was from Invitrogen.

Full length of human PAK4(1-591) was cloned into the pXJ-BirA*-GFP (C) vector with deletion of the stop codon to create pXJ-PAK4-BirA*-GFP[140].

**Western blotting**. Cell lysates were prepared by mixing lysate in 6× protein sample buffer (250 mM Tris-HCl pH 6.8, 6% SDS, 100 mM DTT, 50% glycerol and 0.1% bromophenol blue) and heated at 100 °C for 5 min. Proteins were resolved by SDS-PAGE and transferred to PVDF membranes (Immobilon P) at 100 V for 2 h. These were blocked for 1 h with 10% skimmed milk in phosphate-buffered saline (PBS) +0.3% Tween 20 (also used to dilute antibodies). Membranes were incubated with primary antibodies for 2 h (1:3000), and secondary antibodies 1 h (1:3000) at room temperature. Following extensive washing, bands were detected by Immobilon ECL solution (Millipore).

**Cell-line knockdown with siRNA**. Custom-designed double-stranded siRNAs with the following sequences were used (Horizon Discovery/Dharmacon Inc., Boulder, CO). MDCK (ATCC CRL-2936), control: ON-TARGETplus non-targeting siRNA #1, (antisense) siPAK4: ACUCGCUUCUUCUUCUUCCdTdT and siAfadin: UUUCUCUAGAC-CAUAUUUCdTdT (with mismatches made in the sense strand at 1, 5 and 10 positions to enhance guide strand loading[141]. U2OS, control: AAUUCUCCGAACGUGUCACGU, (sense) siPAK4-1: CCAUGAAGAUGAUUCGGGA and siAfadin: GAAGAAAGAAG-AUUGGAUAUU[37]. Cells were transfected using TransIT X2 (MirusBio) for MDCK cells or Lipofectamine 2000 (Invitrogen) for U2OS, at a final concentration of 50 nM, according to the manufacturers protocol. Cells were fixed between 48 and 72 h after transfection for immuno-staining.

**Generation of stable cell lines**. Human osteosarcoma U2OS (ATCC HTB-96) cells were cultured in high glucose Dulbecco's modified Eagle's media with 4500 mg/L glucose, supplemented with 10% FBS (Hyclone). MDCK cells were cultured in minimum essential medium (MEM) supplemented with 10% FBS, 2 mM L-glutamine, 10 mM sodium pyruvate, 0.15% w/v sodium bicarbonate and 0.1 mM MEM nonessential amino acids. U2OS cells were co-transfected with pXJ-PAK4-BirA*-GFP or control pXJ-BirA*-GFP constructs and pBabe-puro at ratio 20:1 using Lipofectamine 2000 reagent according to the manual provided by supplier. Twenty-four hours later cells were trypsinized, and replated at 10% confluence. Stable cell line selection with 0.8 µg/ml puromycin added 24 h later. When cell colonies were observed at ~7 days, individual clones were isolated and screened by imaging the GFP signal. Positive PAK4 and control cell lines were isolated for affinity protocol or frozen.

**Calcium switch, inhibitor treatment and 3D culture**. MDCK and U2OS cells were grown to 80% confluence on 22 × 22 mm uncoated glass coverslips. Then rinsed in warm calcium-free PBS, and incubated in 5 mM EGTA, serum-free DME for 45 min, until more than 50% cells showed distinct rounding. Full serum (1.8 mM calcium) was then replaced for 45 min to allow basal reattachment before treatment with inhibitor/DMSO in reduced (5%) serum DME for 2–4 h (Blebbistatin 25 µM, Y-27632 10 µM, PF3758309 5 µM).

For 3D culture, $3 \times 10^5$ cells/ml of MDCK cells resuspended in 2% Matrigel media were plated in 100% Matrigel (BD Biosciences) spread on the glass coverslips. Cysts were allowed to grow for 8 days, with media replacement performed every 3 days.

**Fixation, immuno-fluorescence and microscopy**. Cells grown on glass coverslips and Matrigel were fixed in 100% methanol at −20 °C for 5 min after and permeabilized in PBS containing 0.2% TritonX-100 for 10 min. Cells were then blocked with 10% bovine serum albumin (BSA) for 10 min. Primary antibody is incubated for 2 h in PBS containing 5% BSA/0.05% Triton X-100. Specific anti-rabbit or anti-mouse Alexa-488 or Alexa-546-conjugated antibodies (Life Technologies) were diluted 1:200 in the same buffer and incubated at RT in a humidified container. Washed samples were mounted on a microscope slide with AquaPolymount (Polysciences) or Vectashield H-1000 (Vectorlabs).

Wide-field images were acquired using an Axioplan2 microscope equipped with CoolSnap HQ cold CCD camera at ×630 magnification. Confocal images were taken with an inverted Olympus Fluoview FV1000 laser microscope system with a ×60/100 oil lens at 1024 × 1024 pixels, with 0.25 µm Z-stack slices. Image analyses were performed using ImageJ (NIH, USA). Scatter plots and statistical analyses were generated from these data using Prism 9 (GraphPad) software.

**Affinity capture of biotinylated proteins**. Established stable U2OS cell lines expressing PAK4-BirA*-GFP or GFP-BirA* were adapted in either heavy (H) or control light (L) isotopic labelled culture medium for 14 days as previously described[142]. Typically, 4 × 95 mm dishes per cell line were used per experiment. In brief, the cells at ~90% confluence was incubated with 100 µM biotin for 5 h. Each

95 mm dish was lysed in 0.5 ml of 100 mM KCl, 25 mM Tris/HCl pH 8.0, 0.5% Triton-X100, 0.5% DOC, 5 mM EDTA and protease inhibitor cocktail (Roche). The nuclei were removed by centrifugation (16,000 r.c.f., 15 min) and the supernatant fraction made to 0.2% SDS. After sonication the cell lysate was clarified again by centrifugation for 10 min. Neutravidin beads (80 µl slurry) were pre-equilibrated with lysis buffer (in 0.2% SDS) and 2–3 ml of mixed lysate (H and L isotope labelled) incubated with rolling at 4 °C overnight. The beads were then washed twice for 5 min with 25 mM Tris pH 8.0/1% SDS with protease inhibitor cocktail, and then twice with lysis buffer for 10 min at 4 °C. Bound proteins were eluted by adding 1× LDS sample buffer with reducing agent (Novex NuPAGE) and heating (100 °C for 5 min). The eluted proteins were separated on 10% SDS-PAGE and fixed stained with Coomassie blue.

**Stable isotope labelling of amino acids in cell culture (SILAC)**. Stable cell lines were adapted in SILAC media composed of dialysed DMEM (Thermo Fisher Scientific), 10% dialysed FBS (Gibco), 100 U/ml PS, supplemented with amino acids excluding L-arginine and L-lysine. Isotopically labelled 0.4 mM L-arginine and 0.8 mM L-lysine (heavy) or the same concentrations of natural isotopes of L-arginine and L-lysine (light) were added. In each experiment, cells were passaged in H or L media (at least four passages). MS analysis confirmed that efficiency of H isotopic protein labelling was >98%.

**Phospho-proteomic sample preparation and phospho-peptide enrichment**. U2OS were cultured to confluence and PAK inhibitor (5 µM) added for 12 min in medium containing 5% serum. Cells were lysed directly in urea buffer (20 mM HEPES pH 8.0, 9 M urea, 1 mM sodium orthovanadate, 2.5 mM sodium pyrophosphate and 1 mM β-glycerophosphate). The collected lysate was passed through a 26G needle (×5) to shear genomic DNA, centrifuged at 20,000 r.c.f. at 15 °C for 15 min and the supernatant collected and adjusted to 4 mg/ml, and stored at −70 °C. Sequencing-grade trypsin (Promega) was added at 1:50 ratio (w/w) and digestion was carried out for 4 h at 37 °C. Tryptic peptides were reconstituted acetonitrile/0.1% trifluoroacetic acid, and enriched by incubation on iron (Fe3+) IMAC beads by end-over-end rotation for 30 min. The beads were then loaded onto a C18 Stage Tip and eluted onto C18 membranes using 500 mM dibasic sodium phosphate (pH 7.0). Phosphopeptides were eluted with 60 µl 50% acetonitrile/0.1% formic acid, dried and reconstitution in 24 µl of 0.1% formic acid[143].

**Phospho-peptide analysis**. Reconstituted peptides were analysed on an EASY-nLC 1000 (Proxeon, Fisher Scientific) coupled to a Q-Exactive Hybrid Quadrupole-Orbitrap Mass Spectrometer and Tune (2.11 QF1 Build 3006, Thermo Fisher Scientific). Peptides were first trapped onto a C18 pre-column and then separated on a 50 cm analytical column (EASY-Spray Columns, Thermo Fisher Scientific) at 50 °C. A 245 min gradient ranging from 0 to 40% acetonitrile/0.1% formic acid was used. This was followed by a 10 min 2 l gradient ranging from 40 to 80% acetonitrile/0.1% formic acid and remained at 80% acetonitrile/0.1% formic acid for 10 min. Survey full scan MS spectra (m/z 310–2000) were acquired with a resolution of 70k, an AGC target of 3e6, and a maximum injection time of 10 ms. The 20 most intense peptide ions in each survey MS scan with an intensity threshold of 10k, underfill ratio of 1% and a charge state ≥2 were isolated in succession with an isolation window of 2 Th to a target value of 50k, maximum injection time of 50 ms and fragmented by high-energy collision dissociation using a normalized collision energy of 25% in the high-energy collision cell. The MS/MS was acquired with a resolution of 17.5k and a starting m/z of 100. A dynamic exclusion with exclusion duration of 15 s was applied[143]. Raw data were processed by MaxQuant software (v1.5.0.30) involving the built-in Andromeda search engine[144]. Human database searches were performed with tryptic specificity allowing maximum two missed cleavages and three labelled amino acids as well as an initial mass tolerance of 4.5 ppm for precursor ions and 0.5 Da for fragment ions. Labelled arginine and lysine were specified. Cysteine carbamidomethylation was searched as a fixed modification, and N-acetylation, oxidized methionine and phosphorylated serine, threonine and tyrosine were searched as variable modifications. False discovery rates were set to 0.01 for both protein and peptide.

**Protein identification by SILAC MS**. Stained polyacrylamide gel pieces (2 mm) were extracted and digested with trypsin under standard conditions: peptides from each slice were separately subjected to nano-liquid chromatography-MS/MS on Orbitrap or Orbitrap XL (Thermo Fisher). Survey full scan MS spectra (m/z 310–1400) were acquired with a resolution of 60k at m/z 400, an AGC target of 1e6, and a maximum injection time of 500 ms. The 10 most intense peptide ions in each survey scan with an ion intensity of >2000 counts and a charge state ≥2 were isolated sequentially to a target value of 1e4 and fragmented in the linear ion trap by collision-induced dissociation using a normalized collision energy of 35%. A dynamic exclusion was applied using a maximum exclusion list of 500 with one repeat count and exclusion duration of 30 s. Raw data were processed by MaxQuant software (v1.6.0) involving the built-in Andromeda search engine[144]. The searches were performed against the human database as described in the previous section.

**In vitro kinase assay on synthetic PAK4 pseudosubstrate peptide spots**. Peptide spots (PepSPOTs, JPT Peptide Technologies GmbH) contain approximately 5–10 nmol peptide covalently bound to a cellulose-β-alanine membrane as described[108]. The membrane was briefly rehydrated in PBS and blocked with PBS/2% BSA. The kinase reaction (30 °C for 30 min) was carried out in situ with recombinant 100 nM His-PAK4cat[23] and 25 µM ATP (containing 1 µCi γ32P ATP) in 25 mM HEPES pH 7.3, 150 mM KCl, 10 mM MgCl₂, 2 mM dithiothreitol. The membrane was extensively washed in PBS/0.1% SDS buffer and subsequently exposed to X-ray film at −20 °C.

**Reporting summary**. Further information on research design is available in the Nature Research Reporting Summary linked to this article.

## Data availability

All the relevant data have been deposited or are available from the corresponding author upon reasonable request. The mass spectrometry proteomics data have been deposited to the ProteomeXchange Consortium via the PRIDE partner repository with the dataset identifier PXD024464. Phosphosite database (V6.5.9.3) can be found at https://www.phosphosite.org//homeAction.action, Afadin https://www.phosphosite.org/proteinAction.action?id=P55196, Scribble https://www.phosphosite.org/proteinAction.action?id=Q14160, ZO-1 https://www.phosphosite.org/proteinAction.action?id=Q07157, DLG5 https://www.phosphosite.org/proteinAction.action?id=Q8TDM6, and CTNND1 https://www.phosphosite.org/proteinAction.action?id=O60716. Source data are provided with this paper.

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

## Acknowledgements

We thank the Agency for Science, Technology and Research (A*STAR), Singapore for support. Part of this work was funded by the National Medical Research Council Grant NMRC/OFIRG/0067/2018.

## Author contributions

Y.B., F.P.-L.T. and E.M. designed and conducted the experiments and wrote the manuscript. E.Y.W.N. performed the phospho-proteomic study. C.L.F.S., S.W. and J.G carried out the mass-spectroscopy experiment and analyses.

## Competing interests

The authors declare no competing interests.
