## [Peer Review File · Nature Communications]

REVIEWER COMMENTS

Reviewer #1 (Remarks to the Author):

The authors adequately support the hypothesis that PAK4 acts as a component of the Afadin/Nectin junctional sub-compartment. The authors utilized the BioID method with stable isotope labelling to obtain an insight into the cellular environment within 20 nm of PAK4 and identified unique cell-cell junction proteins that were not previously identified by PAK4 pull-down. The authors speculate that Afadin, one of the main hits from BioID, directly or indirectly interacts with PAK4, which was not previously detected using standard protein-protein interaction assays.

More than half of PAK4 proximal proteins were identified by Afadin BioID further strengthening the interaction between the two. Mechanistically, authors show that PAK4 requires prior recruitment of Afadin for its junctional localization and that Afadin is maintained at the site via a PAK4 feedback loop. Cellular PAK4 substrates were identified by investigating phosphor-proteome changes after short treatment with PAK inhibitors. They identified 17 PAK4 sites among the PAK4-proximal proteins. Overall, these data identify PAK4 as a component of the Afadin/Nectin sub-compartment that is distinctly located from tight and adherens junctions.

Major Comments:

1) BioID was used to detect proteins in close proximity to PAK4. Of note, only one of the proteins listed in Figure 1C has been previously described as a substrate of PAK4. Can the authors clarify why they did not observe any previously described PAK4 substrates such as b-catenin, SSH1, etc.? Furthermore, the authors mentioned that there is 0 overlap between PAK4 pull-down data and PAK4 BioID. Can the authors clarify why the direct interactions observed using pull-down were not seen in BioID?

One of the top hits identified using PAK4 pulldown was DHRS2, however it was dismissed as an artifact. Some clarification might be needed to explain why the top hit is an artifact.

2) Considering how important PAK4 localization is for this paper, an IF image comparing PAK4-BirA, PAK4 would help indicate that the addition of the tag did not affect PAK4 localization in the cell.

3) PAK4-BirA cell lines generated showed 4 times higher expression of PAK4 than endogenous. Can the authors clarify why they decided to overexpress, rather than try to control for similar levels of expression?

Minor:

1)“Neither β -catenin nor p120ctn which mark AJs, co-localized with PAK4 in either MDCK or U2OS cells (Figure 3B)” Might be a typo, this was shown in Figure 3A

Reviewer #2 (Remarks to the Author):

Edward Manser discovered for the first time PAK as a serine/threonine protein kinase downstream of Rac and Cdc42 and has been made a major contribution to uncovering its function and mechanism of action. His achievements have been acknowledged all over the world. As part of his

works, Baskaran et al. identified the PAK4-proximal proteins in U2OS human osteosarcoma cells using BioID proximity biotinylation coupled to proteomics and showed that the nectin-afadin system, but not the cadherin-catenin system, is in close proximity to PAK4. Using the same method, they identified the afadin-proximal proteins, revealing the substantial overlap between the two datasets. They further found using U2OS and MDCK cells that PAK4 is localized at cell-cell junctions in an afadin-dependent manner, partly consistent with the previous observations that the junctional localizations of PAK4/Mbt and afadin/canoe are interdependent in *Drosophila*. Finally, they identified 17 PAK4-mediated phosphorylation sites in the PAK4-proximal proteins using phosphoproteomic analysis. The experiments are carefully designed and the data presented in this manuscript are generally convincing. Although the biological meaning(s) of the localization of PAK4 at the nectin-afadin system still remains obscure, the findings of this manuscript are valuable and of broad interest to researchers in this field. This reviewer considers the manuscript to be suitable for publication in Nature Communications after several minor revisions.

Specific comments:

1. The authors should be cautious about using the term “adherens junctions”. For example, the following sentence in page 17 is incorrect:
“In the context of mammalian cells the Afadin/nectin complex is spatially distinct from tight or adherens junctions in several cell types and is likely to contain both specific proteins, and common protein linkers, such as those associated with the F-actin network adjacent to Afadin.”
The term “adherens junctions” is defined as the cell-cell junctions consisting of the nectin-afadin system and the cadherin-catenin system.
2. Insufficient discussion of the results that cadherin, β -catenin, and α -catenin were not included in the PAK4- or afadin-proximal proteins dataset: It has been shown that the nectin-afadin system and the cadherin-catenin system are interconnected (Tachibana et al., 2000), and cooperate to associate the actomyosin with AJs in highly polarized cells (Sakakibara et al., 2020). Given that the BioID method would identify proteins that are located within 20 nm, the above results are somewhat unexpected. One possible explanation is that the cells used in BioID method in the present study and Troyanovsky’s work (Indra et al., 2013) show the interdigitated pattern of AJs and are less polarized. In these cells, the nectin-afadin system and the cadherin-catenin system might be more separated than those of highly polarized cells. It would be better to discuss this issue.
3. Digital image quantification: In Fig. 4C and 4D, the effects of the ROCK inhibition on the junctional localizations of PAK4 and afadin should be analyzed more carefully. The localizations of PAK4 and afadin at the tricellular junction appear to be enhanced by the ROCK inhibition, while the enhancement of those at the bicellular junction does not appear so prominent. It would be better to quantify the signal intensities at the bicellular junction and the tricellular junction separately and provide the quantification procedure in detail. Besides the above, this reviewer felt that the relation between PAK4 and ROCK is beyond the scope of the present manuscript.
4. Presentation of data: It would be better to provide the raw data of BioID list of the afadin-proximal proteins in table t1 as shown for that of PAK4.
5. Presentation of figures: It would be better to place the schematic, which is now on the fig. 5, on the last figure.
6. Reference: Some key publications are missing regarding the identification of afadin (Mandai et al., *J Cell Biol*, 139(2):517-28, 1997), the role of nectins in spermatogenesis (Ozaki-Kuroda et al., *Curr Biol*, 12(13):1145-50, 2002), the activation of cdc42 by nectins (Fukuhara et al., *J Cell Biol*, 166(3):393-405, 2004), the role of the afadin-p120ctn interaction in the regulation of E-cadherin endocytosis (Hoshino et al., *J Biol Chem*, 280(25):24095-103, 2005), and the nectin-based microcluster (Mizutani and Takai, *Biochem J*, 473(18):2691-715, 2016).

Reviewer #3 (Remarks to the Author):

In their study Baskaran et al use an elegant combination of different proximity proteomics, fluorescence microscopy, siRNA genetic perturbation and phosphoproteome analyses to map the local proteome environment of the kinase PAK4 which localizes at cell-cell junctions. BirA* proximity proteomics experiments for ectopically expressed PAK4 identify many cell junction proteins including afadin and nectin using a simple SILAC fold-change cut off approach. Reciprocal afadin-BirA* experiments confirm the junctional proximity of PAK4, afadin alongside 11 other proteins. From a technical standpoint it is interesting to see here that conventional GFP pulldowns of PAK4-BirA*-GFP do not yield any of the interesting junctional proteins, most likely since the cell junctions cannot be solubilized and immunoprecipitated, highlighting the advantage of the BioID approach. The authors use then a phosphoproteome approach with a pan-PAK inhibitor (PF3758309) in comparison to a PAK1/2/3 but not PAK4 inhibitor (Frax597) and identify 16 candidate PAK4 substrate phosphosites (Fig 6b) on other junctional proteins but not afadin or nectin. In vitro kinase assays with 32 synthetic peptides confirm 17 phosphosites on 5 substrate phosphoproteins, including DLG5 and CTNND1 with strong signals and afadin at pS1083 with weak signal. To study co-regulatory effects on junctional localization the authors use immunofluorescence techniques and siRNA-mediated genetic perturbations and find that PAK4 requires afadin to localize to cell-cell junctions.

Major Points:

- While the authors show a regulatory effect of afadin on PAK4 localization to cell junctions, no phenotypic information on cell junction formation in cell lines, animal models, or healthy or diseased tissues are provided. The authors provide lists of new proteins that co-localize with PAK4 and are potential substrates of the kinase, but do not provide further evidence how these functional interactions affect physiological or pathophysiological processes
- For all proteomics analysis (see Figures 1, 2, 6) the authors use simple SILAC fold change cut-offs and do not indicate at all how reproducible their data was in between biological replicates. The authors need to run statistical tests to address reproducibility and also correct resulting p-values for multiple testing to address what biotinylated or phosphorylated proteins reach significance.

Minor Points:

- Figure 1B: PAK4-BirA-GFP expression is >10x stronger than endogenous PAK4 expression. The authors need to show GFP images of where the ectopic PAK4-BirA-GFP protein localizes in the cell. Is this similar to the localization of endogenous PAK4? If not, what background biotinylation of non-junctional proteins was observed?
- Figure 1 and Figure 6: The authors should explain why no label swap SILAC replicates were performed to distinguish SILAC labeling artefacts from true up-regulated proteins
- Did the authors search for biotinylated peptides in their BioID data?
- The authors should state in the text the reason why for some figures/analyses U2Os cells were chosen and for other MDCK. In the current story line the cell lines seem interchangeable, which may not be the case for some of the findings
- Page 18: It would be useful to indicate the SILAC labeling efficiency and arginine to proline conversion rate of SILAC labeled cells

Summary

We thank the reviewers for their perceptive comments.

Based on additional experiments and analyses, we are pleased to return a new version of manuscript. The marked up text indicates the substantial changes in this version (in red) as appended. The new data in response to your various requests can be found in main Figures 3B, 4B 5C/D and Figure 8.

To demonstrate that our BioID analysis uncovers new proteins linked to the afadin complex we have characterized LZTS2 (ranked 3 in our PAK BioID / Figure 1), which has not been linked for cell-cell junctions previously. This 'leucine zipper tumour suppressor' protein LZTS2 is an N-myristoylated protein of unknown function. In mice LZTS2 knockout leads to kidney dysfunction (Peng et al., 2011). The new data (Fig 4B) shows that LZTS2 protein is indeed localized at cell-cell junctions. The junctional LZTS2 is more closely aligned to Afadin than bCatenin consistent with the notion that LZTS2 is part of the Afadin complex (see Fig 2D). This is added to our model (figure 8) while work on other new components is ongoing it depends on high quality antibodies.

We believe these changes comprehensively address the issues that were highlighted with a full rebuttal to the points raised given below.

The proteomic data can be accessed at ProteomeXchange as follows:

accession: PXD024464 **Username:** reviewer_pxd024464@ebi.ac.uk

Password: S0STFRMp

Reviewer #1 (Remarks to the Author):

The authors adequately support the hypothesis that PAK4 acts as a component of the Afadin/Nectin junctional sub-compartment. The authors utilized the BioID method with stable isotope labelling to obtain an insight into the cellular environment within 20 nm of PAK4 and identified unique cell-cell junction proteins that were not previously identified by PAK4 pull-down. The authors speculate that Afadin, one of the main hits from BioID, directly or indirectly interacts with PAK4, which was not previously detected using standard protein-protein interaction assays.

More than half of PAK4 proximal proteins were identified by Afadin BioID further strengthening the interaction between the two. Mechanistically, authors show that PAK4 requires prior recruitment of Afadin for its junctional localization and that Afadin is maintained at the site via a PAK4 feedback loop. Cellular PAK4 substrates were identified by investigating phosphor-proteome changes after short treatment with PAK inhibitors. They identified 17 PAK4 sites among the PAK4-proximal proteins. Overall, these data identify PAK4 as a component of the Afadin/Nectin sub-compartment that is distinctly located from tight and adherens junctions.

Major Comments:

1) BioID was used to detect proteins in close proximity to PAK4. Of note, only one of the proteins listed in Figure 1C has been previously described as a substrate of PAK4. Can the authors clarify why they did not observe any previously described PAK4 substrates such as b-catenin, SSH1, etc.?

Response. In considering the above issue we anticipate that some substrates may interact via a 'kiss and run' mechanism, or that some of the existing described substrates are not physiologically relevant in the context of U2OS cells. We have nonetheless extracted peptide data that relates to additional published PAK4 substrates namely ARHGEF11, STMN and Raf-1 (see updated Table T3). The phospho-peptide data for

bCat/p120cat /Bad/GEFH1 was already provided. The p-peptide enrichment in our protocol MS may not necessarily include sites previously 'identified in such 'PAK4 targets'. This is likely as 'candidate approaches' and low throughput methods (for example by Ser->Ala mutagenesis or phosphorylation *in vitro*) can uncover peptides that have low stoichiometry of phosphorylation, or are sites in peptides not amenable to tryptic /MS analysis. These issues are discussed in more detail on p11, para 2).

Furthermore, the authors mentioned that there is 0 overlap between PAK4 pull-down data and PAK4 BioID. Can the authors clarify why the direct interactions observed using pull-down were not seen in BioID? One of the top hits identified using PAK4 pulldown was DHRS2, however it was dismissed as an artifact. Some clarification might be needed to explain why the top hit is an artifact.

Response. We believe the interaction between PAK4 and proximal proteins are relatively weak in bulk solution. In our own PD experiment described in this paper using PAK4-GFP expressed in stable U2OS cells, only DHRS2 was enriched relative to control (p 5). This protein is annotated as mitochondrial Hep27, which contains a typical import leader sequence (Deisenroth et al., 2010). Once it is released by detergent DHRS2 may interact with PAK4. We have not explicitly tested DHRS2 because of this intra-mitochondrial location. In experiments where Flag-PAK4 was transiently over-expressed (Zhao et al., 2017) the 313 'PAK4-associated proteins' obtained from epithelial MCF7 found by MS may be the result of the need for better controls (described on p6). The reports of GEFH1 and (Callow et al., 2005) and ARHGEF11 (Barac et al., 2004) as detected by yeast-2 hybrid with PAK4 may reflect the higher sensitivity of this method.

2) Considering how important PAK4 localization is for this paper, an IF image comparing **PAK4-BirA, and PAK4** would help indicate that the addition of the tag did not affect PAK4 localization in the cell.

We provide new data showing localization of PAK4-BirA in U2OS cells (Fig S1).

3) PAK4-BirA cell lines generated showed 4 times higher expression of PAK4 than endogenous. Can the authors clarify why they decided to overexpress, rather than try to control for similar levels of expression?

Protein kinases are less abundant than typical junctional proteins and PAK4 levels are 1/10 of PAK2. The PAK4-BirA-GFP cell lines were identified by visual screening of U2OS colonies (after fixation) and only weakly detectable (fluorescence with oil).

Minor:

1)“Neither β -catenin nor p120ctn which mark AJs, co-localized with PAK4 in either MDCK or U2OS cells (Figure 3B)” Might be a typo, this was shown in Figure 3A

We have removed this statement.

Reviewer #2 (Remarks to the Author):

Edward Manser discovered for the first time PAK as a serine/threonine protein kinase downstream of Rac and Cdc42 and has been made a major contribution to uncovering its function and mechanism of action. His achievements have been acknowledged all over the world. As part of his works, Baskaran et al. identified the PAK4-proximal proteins in U2OS human osteosarcoma cells using BioID proximity biotinylation coupled to proteomics and showed that the nectin-afadin system, but not the cadherin-catenin system, is in close proximity to PAK4. Using the same method, they identified the afadin-proximal proteins, revealing the substantial overlap between the two datasets. They further found using U2OS and MDCK cells that PAK4 is localized at cell-cell junctions in an afadin-dependent manner, partly consistent with the previous observations that the junctional localizations of PAK4/Mbt and afadin/canoe are interdependent in *Drosophila*. Finally, they identified 17 PAK4-mediated phosphorylation sites in the PAK4-proximal proteins using phosphoproteomic analysis. The experiments are carefully designed and the data presented in this manuscript are generally convincing. Although the biological meaning(s) of the localization of PAK4 at the nectin-afadin system still remains obscure, the findings of this manuscript are valuable and of broad interest to researchers in this field.

This reviewer considers the manuscript to be suitable for publication in Nature Communications after several minor revisions.

Specific comments:

1. The authors should be cautious about using the term “adherens junctions”. For example, the following sentence in page 17 is incorrect:

“In the context of mammalian cells the Afadin/nectin complex is spatially distinct from tight or adherens junctions in several cell types and is likely to contain both specific proteins, and common protein linkers, such as those associated with the F-actin network adjacent to Afadin.”

The term “adherens junctions” is defined as the cell-cell junctions consisting of the nectin-afadin system and the cadherin-catenin system.

Response. We are grateful for the insight, and agree that the current terminology is not consistent. Thus we take the reviewers advice and try to differentiate throughout the cadherin-based adhesions versus nectin-based adhesions (rather than using the term AJ). Depending on context may or may not be spatially segregated. These changes have been made throughout.

2. Insufficient discussion of the results that cadherin, β -catenin, and α -catenin were not included in the PAK4- or afadin-proximal proteins dataset: It has been shown that the nectin-afadin system and the cadherin-catenin system are interconnected (Tachibana et al., 2000), and cooperate to associate the actomyosin with AJs in highly polarized cells (Sakakibara et al., 2020). Given that the BioID method would identify proteins that are located within 20 nm, the above results are somewhat unexpected. One possible explanation is that the cells used in BioID method in the present study and Troyanovsky’s work (Indra et al., 2013) show the interdigitated pattern of AJs and are less polarized. In these cells, the nectin-afadin system

and the cadherin-catenin system might be more separated than those of highly polarized cells. It would be better to discuss this issue.

Agreed. To address the above we have now provided additional data on U2OS cells (which exhibit interdigitated adhesions) in the new Figure 4. We also look at immature MDCK adhesions when cells are at low density, and find that the cadherin-rich lateral puncta have relatively low levels of Afadin /PAK4 (new Fig 3B). The early papers using cadherin-based BioID (Van Itallie 2014, and Guo 2014) did not identify Afadin as highly enriched.

Since Afadin is an abundant adaptor (much like p120 Cat) we anticipate it should be more enriched than we observe, if nectin and cadherin are intermingled. It seems from our data and the above, that these two cell-cell junction proteins are segregated, at least at the nano-scale, and sometimes at the meso-scale (cf. Figure 3).

3. Digital image quantification: In Fig. 4C and 4D, the effects of the ROCK inhibition on the junctional localizations of PAK4 and afadin should be analyzed more carefully. The localizations of PAK4 and afadin at the tricellular junction appear to be enhanced by the ROCK inhibition, while the enhancement of those at the bicellular junction does not appear so prominent. It would be better to quantify the signal intensities at the bicellular junction and the tricellular junction separately and provide the quantification procedure in detail.

Besides the above, this reviewer felt that the relation between PAK4 and ROCK is beyond the scope of the present manuscript.

As requested we have repeated this experiment and investigated both bi-cellular and tricellular junctions (cf Figure 5 CD). Certainly this observation is of interest with respect to mechanism as The ROCK adaptor Shroom is PAK4/Afadin proximal and elevated Afadin leads to ROCK-dependent increase in actin stress fibres (Saito, 2015).

4. Presentation of data: It would be better to provide the raw data of BioID list of the afadin-proximal proteins in table t1 as shown for that of PAK4.

We have expanded the Afadin-proximal list as requested (in updated Table S1)

5. Presentation of figures: It would be better to place the schematic, which is now on the fig. 5, on the last figure.

We agree it would be neater to have a model in the last figure. The original rationale for the placing in Figure 5 was to aid interpretation of Results. We have now moved the model to Figure 8.

6. Reference: Some key publications are missing regarding the identification of afadin (Mandai et al., J Cell Biol, 139(2):517-28, 1997), the role of nectins in spermatogenesis (Ozaki-Kuroda et al., Curr Biol, 12(13):1145-50, 2002), the activation of Cdc42 by nectins (Fukuhara et al., J Cell Biol, 166(3):393-405, 2004), the role of the afadin-p120ctn interaction in the regulation of E-cadherin endocytosis (Hoshino et al., J Biol Chem, 280(25):24095-103, 2005), and the nectin-based microcluster (Mizutani and Takai, Biochem J, 473(18):2691-715, 2016).

Thank you for pointing us to these additional publications which are required to properly cover the Afadin field. We have now added Mandai 1997, Ozaki-Kuroda 2002, Fukuhara 2004 in the Introduction. Mizutani & Takai 2016 are added in Results section on Page 7 in discussing punctate junctions.

Reviewer #3 (Remarks to the Author):

In their study Baskaran et al use an elegant combination of different proximity proteomics, fluorescence microscopy, siRNA genetic perturbation and phosphoproteome analyses to map the local proteome environment of the kinase PAK4 which localizes at cell-cell junctions. BirA* proximity proteomics experiments for ectopically expressed PAK4 identify many cell junction proteins including afadin and nectin using a simple SILAC fold-change cut off approach.

Reciprocal afadin-BirA* experiments confirm the junctional proximity of PAK4, afadin alongside 11 other proteins. From a technical standpoint it is interesting to see here that conventional GFP pulldowns of PAK4-BirA*-GFP do not yield any of the interesting junctional proteins, most likely since the cell junctions cannot be solubilized and immunoprecipitated, highlighting the advantage of the BioID approach. The authors use then a phosphoproteome approach with a pan-PAK inhibitor (PF3758309) in comparison to a PAK1/2/3 but not PAK4 inhibitor (Frax597) and identify 16 candidate PAK4 substrate phosphosites (Fig 6b) on other junctional proteins but not afadin or nectin. In vitro kinase assays with 32 synthetic peptides confirm 17 phosphosites on 5 substrate phosphoproteins, including DLG5 and CTNND1 with strong signals and afadin at pS1083 with weak signal. To study co-regulatory effects on junctional localization the authors use immunofluorescence techniques and siRNA-mediated genetic perturbations and find that PAK4 requires afadin to localize to cell-cell junctions.

Major Points:

(1) While the authors show a regulatory effect of afadin on PAK4 localization to cell junctions, no phenotypic information on cell junction formation in cell lines, animal models, or healthy or diseased tissues are provided. The authors provide lists of new proteins that co-localize with PAK4 and are potential substrates of the kinase, but do not provide further evidence how these functional interactions affect physiological or pathophysiological processes.

Response. The reviewer is right - the role of *Drosophila* PAK4 (Mbt) in epithelial polarization is better characterized than in mammalian cells and tissues. Nonetheless several studies implicate PAK4 in patho-physiological processes. We have provided these examples in the updated text. For example, endothelial cell lumen and tube formation which is essential for vasculature development and repair requires PAK4 (Koh et al. 2007), explaining the lack of vasculature throughout the extraembryonic tissue in PAK4 KO mice (Tian et al. 2009). Our findings that PAK4 is closely linked to Afadin in mammalian provides a platform to understand interplay between these important effectors of Rap1 and Cdc42. We contend that MDCK represents a well established cell model that reflects many aspects of epithelia organization behaviour *in vivo*.

Cdc42 is one of the most highly conserved polarity organizers that promotes an epithelial apical/basal axis, and generates a stable epithelium. It is notable that a comprehensive screen of Cdc42 effectors found only PAK4 and Par6B to promote the assembly of apical junctions in human bronchial epithelial cells (Wallace 2010; Jin 2015). Similarly in 3D culture, spheroid formation is disrupted by knockdown of PAK4 (Koh et al., 2008) or afadin (Gao et al., 2017).

The novelty of our work is to emphasize that the afadin compartment is biochemically distinct from the cadherin one, matching by optical super-resolution imaging (Indra 2013). This work adds to that of our colleagues (Tan et al. 2020) using APEX2 to develop a proteome of the apical-lateral border in which components and apical and lateral compartment markers were spatially resolved. The Crumbs complex (Pals1, PatJ, Lin7c, and Crumbs3) was defined a spatially distinct compartment apical of tight junctions, and coined the vertebrate marginal zone (VMZ).

Does BioID identify a physiologically relevant compartment? We decided to test the functional localization of LZTS2, which is the most abundant of the LZTS proteins to be expressed most commonly used cells lines (including U2OS). Previous work on LZTS proteins and their partners SPAR (SIPA1L1) has focussed on their co-localization in synapses (where Afadin is also important). LZTS2 knockout mice have defects in the kidney indicating an important role outside the CNS (Peng et al. 2011 J Biol Chem. 286: 40331-42). In the new Figure 4 we show for the first time that LZTS2 is junctionally enriched, and that colocalizes with Afadin but to a lesser extent with b-cat. This new work provides evidence for the notion of local network of afadin-associated proteins in cell junctions.

(2) For all proteomics analysis (see Figures 1, 2, 6) the authors use simple SILAC fold change cut-offs and do not indicate at all how reproducible their data was in between biological replicates. The authors need to run statistical tests to address reproducibility and also correct resulting p-values for multiple testing to address what biotinylated or phosphorylated proteins reach significance.

Response. We now add the details requested to the tables. The BioID data for PAK4 and afadin were combined from two independent experiments. For technical reasons both were carried out using heavy labelling (H) of PAK4 BirA-GFP cells - as protein contaminants and serum-derived proteins are light (L). Specifically the standard deviation value for each protein derived from all SILAC informative peptide pairs is provided. These errors are generated within the Maxquant software as described (Cox & Mann 2008).

For the p-peptide enrichment and quantitation experiments were carried out in 'forward' and 'reverse' (cf. control-H/ Frax597-L and Control-L/ Frax597-H). In this case the SEM value is provided based on 3 or more measurements (MS peptide pairs). Here the p-peptide SILAC ratio is usually based on fewer data points than for the BioID enrichment.

Minor Points:

Figure 1B: PAK4-BirA-GFP expression is >10x stronger than endogenous PAK4 expression. The authors need to show GFP images of where the ectopic PAK4-BirA-GFP protein localizes in the cell. Is this similar to the localization of endogenous PAK4? If not, what background biotinylation of non-junctional proteins was observed?

Response. We now provide images of the PAK4-BirA-GFP localization as requested (Figure S1).

-Figure 1 and Figure 6: The authors should explain why no label swap SILAC replicates were performed to distinguish SILAC labeling artefacts from true up-regulated proteins.

These reasons are described above (Point 2).

- Did the authors search for biotinylated peptides in their BioID data?

Response. We are aware that identification of biotinylated peptides requires some modification of the search algorithm, for example Renuse et al. 2020. However the level of (exogenous protein) labelling with the BirA(R112G) is generally insufficient (cf. PAK4-BirA). We are currently trying to implement biotinylated lysines using the TurboID enzyme.

- The authors should state in the text the reason why for some figures/analyses U2Os cells were chosen and for other MDCK. In the current story line the cell lines seem interchangeable, which may not be the case for some of the findings

Response. We originally validated the BioID method in U2OS cells (Dong et al., 2016) and discovered that Kindlin-2 is enriched in cell-cell junctions in addition to focal adhesions. Here we have investigated in parallel PAK4 and Afadin in U2OS, which have 'brush-like' adhesions, and MDCK cells which have more differentiated junctional phenotype (although many commercial antibodies do not work on dog cells). A substantial number of studies have looked at Afadin in MDCK cells (for example Sato 2006, Ooshio et al. 2010, Choi et al. 2016).

- Page 18: It would be useful to indicate the SILAC labelling efficiency and arginine to proline conversion rate of SILAC labeled cells

We have noted this on page 20 of MS.

REVIEWERS' COMMENTS

Reviewer #1 (Remarks to the Author):

The authors have addressed my concerns

Reviewer #2 (Remarks to the Author):

The authors have fully addressed to my concerns. I recommend that this manuscript is now suitable for publication in Nature Communications.

Reviewer #4 (Remarks to the Author):

Comment from the Editor:

"As previously mentioned, we would not need a full review in this case but rather your assessment of our previous Reviewer (#3)'s questions which are pasted below."

Reviewer Q1: R3 is asking for further evidence/experiments on how the newly discovered PAK4 interactors functionally affect physiological or pathophysiological processes.

Response 1: Baskaran et al. add information from the literature about the proteins found and add one experiment (new Figure 4) showing evidence that LZTS2 is indeed junctionally enriched. Baskaran et al. argue further that the main goal of this manuscript is to show that afadin compartment is distinct from the cadherin compartment, implying that the functional relevance of newly identified interactions is outside the scope of the current manuscript.

Assessment 1: The authors do not formally provide R3 with the data requested, e.g. evidence how PAK4, afadin and the newly found functional interactions affect physiological or pathophysiological processes. Editorial decision if further functional evidence is required for publication in NC or not. However, having stated this - from this reviewer's perspective - this was not the main goal of the authors. The authors provide evidence that PAK4 is a component of the Afadin/Nectin sub-compartment which is distinctly located from tight and adherens junctions. Afadin was further shown to have a regulatory effect on PAK4 localization. Further functional validations would be always "nice to have", but would require a substantial experimental effort which is outside the scope of the current, interesting manuscript.

Reviewer Q2: Need for statistical tests

Response 2: MaxQuant analysis data (SEM & SD) and is provided now on the tables

Assessment 2: o.k.

Minor Points: o.k.

REVIEWERS' COMMENTS

Reviewer #1 (Remarks to the Author):

The authors have addressed my concerns

Reviewer #2 (Remarks to the Author):

The authors have fully addressed to my concerns. I recommend that this manuscript is now suitable for publication in Nature Communications.

Reviewer #4 (Remarks to the Author):

Comment from the Editor:

"As previously mentioned, we would not need a full review in this case but rather your assessment of our previous Reviewer (#3)'s questions which are pasted below."

Reviewer Q1: R3 is asking for further evidence/experiments on how the newly discovered PAK4 interactors functionally affect physiological or patho-physiological processes.

Response 1: Baskaran et al. add information from the literature about the proteins found and add one experiment (new Figure 4) showing evidence that LZTS2 is indeed junctionally enriched. Baskaran et al. argue further that the main goal of this manuscript is to show that afadin compartment is distinct from the cadherin compartment, implying that the functional relevance of newly identified interactions is outside the scope of the current manuscript.

Assessment 1: The authors do not formally provide R3 with the data requested, e.g. evidence how PAK4, afadin and the newly found functional interactions affect physiological or pathophysiological processes. Editorial decision if further functional evidence is required for publication in NC or not. However, having stated this - from this reviewer's perspective - this was not the main goal of the authors. The authors provide evidence that PAK4 is a component of the Afadin/Nectin sub-compartment which is distinctly located from tight and adherens junctions. Afadin was further shown to have a regulatory effect on PAK4 localization. Further functional validations would be always "nice to have", but would require a substantial experimental effort which is outside the scope of the current, interesting manuscript.

Author comment

As noted above we have provided data overall to support the claim that the afadin/PAK4 compartment is (in the contexts discussed) distinct from cadherin (AJ) and tight junction (TJ) structures. Understanding the mechanisms that keep these various junctional compartments segregated will require the underlying protein-protein interactions.

Reviewer Q2: Need for statistical tests

Response 2: MaxQuant analysis data (SEM & SD) and is provided now on the tables

Assessment 2: o.k.

Minor Points: o.k.